# EnsembleVLA: Ensemble Learning for Vision-Language Action Models

**Mingchen Song** [1 2]  **Xiang Deng** [† 1 3]  **Jie Wei** [1]  **Dongmei Jiang** [† 2]  **Liqiang Nie** [1]  **Weili Guan** [† 1 3]

## Abstract

Diverse Vision-language-action (VLA) models have been proposed and demonstrated remarkable capabilities in robotic manipulation. However, how to effectively ensemble VLAs to further enhance performance remains largely unexplored, as conventional ensemble techniques designed for discriminative tasks cannot be directly applied to generative action policies with high-dimensional, multimodal distributions. To address this challenge, we propose **EnsembleVLA**, an energy-based framework that enables principled ensemble of diverse VLA models. We establish a unified theoretical framework showing that both diffusion-based and flow-based VLA models can be formulated as energy-based models, where additive energy combination naturally induces policy composition at the distribution level. This theoretical foundation enables multiple pre-trained policies to be seamlessly aggregated into a stronger ensemble policy. Building upon this compositional framework, EnsembleVLA further incorporates learnable composition weights for dynamic policy balancing, coupled with a confidence-aware gating mechanism that adaptively modulates bounded residual corrections, collectively ensuring stable and robust task execution. Extensive experiments demonstrate that EnsembleVLA achieves competitive performance across various tasks in both simulated and real-world environments.

## 1. Introduction

*Two heads are better than one.*

*— English proverb*

[†]Corresponding Author. [1]Harbin Institute of Technology (Shenzhen), Shenzhen, China [2]PengCheng Laboratory, Shenzhen, China [3]Shenzhen Loop Area Institute, Shenzhen, China. Correspondence to: Xiang Deng <dengxiang@hit.edu.cn>, Dongmei Jiang <jiangdm@pcl.ac.cn>, Weili Guan <honeyguan@gmail.com>.

*Proceedings of the $43^{rd}$ International Conference on Machine Learning*, Seoul, South Korea. PMLR 306, 2026. Copyright 2026 by the author(s).

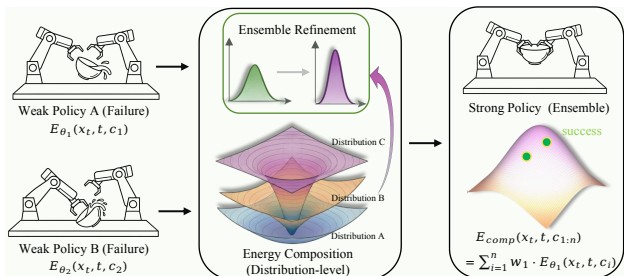

*Figure 1.* **Energy-Based Ensemble Policy.** To compose diverse weak policies into a stronger policy, we first establish a unified energy-based framework to ensemble VLAs at the distribution level, and further incorporate ensemble refinement to refine the composed actions.

Vision-Language-Action (VLA) models represent a transformative paradigm in embodied AI (Qu et al., 2025; Wen et al., 2025; Kim et al., 2024; Song et al., 2025a), enabling robotic systems to jointly reason over visual perception, language understanding, and physical action generation. Methods such as DP (Chi et al., 2025), DP3 (Ze et al., 2024), OpenVLA-oft (Kim et al., 2025), $\pi_0$ (Black et al., 2024), and $\pi_{0.5}$ (Intelligence et al., 2025) have emerged as dominant frameworks by leveraging robot demonstration data and advanced generative backbones. Compared to conventional modular pipelines that decouple perception, planning, and control (Hermans et al., 2013), VLA models demonstrate remarkable proficiency in understanding complex multimodal environments and executing precise actions through end-to-end learning. Their capabilities span a broad spectrum of tasks, from elementary pick-and-place (Jiang et al., 2023) operations to intricate long-horizon manipulation requiring sophisticated reasoning (Li et al., 2025a; Zhao et al., 2025; Xian et al., 2023).

Although current VLA models have demonstrated remarkable capabilities in robotic manipulation (Duan et al., 2024; Li et al., 2024; Team et al., 2024; Wang et al., 2025; Zhang et al., 2025b; Li et al., 2023; Cen et al., 2025; Ma et al., 2024; Yang et al., 2025; Song et al., 2026), they still exhibit performance limitations on certain challenging tasks. This limitation motivates an interesting question: *Can we ensemble multiple VLA models to achieve capabilities beyond any single model?* Ensemble learning (Yang et al., 2023; Dong et al., 2020), which aggregates predictions from multiple

models, has proven highly effective in discriminative tasks such as image classification. It offers a principled framework for leveraging the complementary strengths of diverse models. However, extending ensemble methods to VLAs remains non-trivial, primarily due to the high-dimensional action space and the inherently multimodal nature of action distributions.

Motivated by this limitation, we draw inspiration from Energy-Based Models (EBMs) (LeCun et al., 2006; Hinton, 2002) and propose **EnsembleVLA**. As illustrated in Figure 1, our fundamental insight is to formulate individual VLA policies as independent energy functions. Grounded in EBMs theory, we observe that the summation of energy functions mathematically corresponds to the product of their underlying probability distributions (Du et al., 2020; 2023; Guo et al., 2023; Yoon et al., 2024). This perspective enables us to systematically synthesize multiple policies into an ensemble policy that effectively captures the complementary strengths of each base policy.

Specifically, we first establish an elegant theoretical unification of diffusion-based and flow-based VLA models. Although these two paradigms operate through distinct generative mechanisms, where diffusion models rely on stochastic differential equations (Song et al., 2020b; Zhao et al., 2022) and flow matching employs deterministic ordinary differential equations (Lipman et al., 2022; Li et al., 2025b), we derive that both can be formulated within a unified energy-based framework. Under this framework, the score function in diffusion models and the velocity field in flow matching both correspond to the negative gradient of an energy function. This theoretical equivalence provides a principled foundation for composing multiple policies at the distribution level, thereby enabling principled ensembling of diverse pre-trained VLA models.

Building upon the composed action distribution, we further refine the composed actions through two complementary mechanisms. First, we introduce learnable composition weights that adaptively assign importance to base policies, enabling flexible policy balancing based on task requirements. Second, we design a confidence-aware gating mechanism consisting of a Delta-Net that predicts fine-grained residual corrections and a Gate-Net that estimates refinement confidence. To ensure stable and robust task execution, we introduce a conservative optimization strategy that selectively activates bounded residual corrections guided by confidence-aware gating signals. This design enables the ensemble to leverage complementary strengths of individual base policies while compensating for their weaknesses.

Our main contributions can be summarized as follows:

- To unlock the potential of combining diverse pretrained VLA models, we propose EnsembleVLA, a novel framework grounded in ensemble learning principles. By treating diverse pretrained VLA models as base learners, our approach leverages energy-based model theory to aggregate them into a stronger, unified policy.

- To enable principled composition across diffusion-based and flow-based VLAs, we establish an elegant theoretical unification of the two paradigms. By demonstrating the mathematical connections among score functions, velocity fields, and energy gradients, we provide a foundation for ensembling diverse VLA models within a unified energy-based framework.

- To further refine the composed actions, EnsembleVLA employs learnable weighting coefficients to dynamically balance the contribution of each base policy, along with a confidence-aware gating module for bounded residual refinements, collectively ensuring stable and robust task execution.

- We conduct comprehensive experiments in both the RoboTwin2 simulation environment and real-world scenarios. The results demonstrate that our framework effectively ensembles different base policies, achieving state-of-the-art performance across a wide variety of manipulation tasks.

## 2. Related Work

### 2.1. Vision Language Action Models

Vision-Language-Action (VLA) models (Shukor et al., 2025; Li et al., 2026; Liu et al., 2024a; Zitkovich et al., 2023; Liu et al., 2024b; Bi et al., 2025; Liu et al., 2025; Faroni et al., 2025; Sapkota et al., 2025; Song et al., 2025b) have achieved significant progress in robotic tasks. These models enable robots to interpret high-level instructions expressed in natural language and execute corresponding actions in complex, real-world environments. Early representative work such as RT-2 (Zitkovich et al., 2023) and OpenVLA (Kim et al., 2024) established strong baselines in multi-task learning and cross-task generalization. Building on this foundation, recent work has diverged into three main technical directions. First, some approaches refine the base policy learning pipeline through mechanisms such as parallel decoding and action chunking (Kim et al., 2025; Zhao et al., 2024), achieving smoother control with continuous action spaces. Second, others formulate robot policy as conditional denoising diffusion processes (Ze et al., 2024; Chi et al., 2025), enabling diverse and adaptive behavior generation. Third, complementary methods integrate flow matching architectures with pre-trained Vision-Language Models (Black et al., 2024; Intelligence et al., 2025), directly leveraging internet-scale semantic knowledge. However, existing VLA methods develop independently, each excelling in different aspects yet lacking a principled mechanism to

combine them. In contrast, our EnsembleVLA proposes an energy-based framework that enables seamless ensemble of diverse VLA policies through additive energy composition.

## 2.2. Ensemble Learning

Ensemble learning (Yang et al., 2023; Reuss et al., 2023; Gkanatsios et al., 2023; Wang et al., 2024; Cao et al., 2025) combines multiple models to achieve better predictive performance. Classical ensemble methods, including bagging (Breiman, 1996), boosting (Freund & Schapire, 1997), and stacking (Wolpert, 1992), have been widely used in enhancing predictive performance and robustness for discriminative tasks such as classification. In contrast, EnsembleVLA proposes a unified framework that leverages ensemble learning principles for VLA models composition by treating diffusion and flow-based paradigms as EBMs to enable principled distribution-level composition. Our framework establishes a learnable ensemble mechanism that integrates adaptive weight balancing and a confidence-aware gating strategy to selectively refine composed actions across multiple VLA policies.

## 2.3. Energy-Based Models

EBMs (LeCun et al., 2006; Zhang et al., 2025a; Liu et al., 2022; Nie et al., 2021) define probability distributions through learnable energy functions $p_\theta(x) \propto \exp(-E_\theta(x))$, where lower energy indicates higher probability. A fundamental property is that the score function equals the negative energy gradient: $\nabla_x \log p_\theta(x) = -\nabla_x E_\theta(x)$, which connects EBMs to score-based diffusion models (Du & Mordatch, 2019; Song et al., 2020b) and flow matching (Balcerak et al., 2025). A key advantage of EBMs is their natural support for compositional generation: additive energy combination corresponds to multiplying the underlying distributions (Du et al., 2020). In contrast to prior work, our EnsembleVLA establishes a unified energy-based theoretical framework that encompasses both diffusion and flow-based VLA policies, leveraging the compositional property of EBMs to enable principled ensemble of diverse generative policies to achieve a stronger ensemble policy.

## 3. Method

### 3.1. Preliminary

**Diffusion Models.** Diffusion models (Ho et al., 2020; Song et al., 2020b) provide a powerful framework for generative modeling by learning to reverse a gradual noising process. The forward process progressively corrupts data into noise and is defined as a stochastic differential equation (SDE) over $t \in [0, T]$:

$$\mathrm{d}\mathbf{x}_t = f(\mathbf{x}_t, t)\,\mathrm{d}t + g(t)\,\mathrm{d}W_t, \tag{1}$$

where $W_t$ denotes the standard Wiener process, and $f(\mathbf{x}_t, t)$, $g(t)$ represent the drift and diffusion coefficients governing the noise injection schedule. By Anderson's theorem (Anderson, 1982), the reverse-time SDE is:

$$\mathrm{d}\mathbf{x}_t = \left[ f(\mathbf{x}_t, t) - g(t)^2 \nabla_{\mathbf{x}_t} \log p_t(\mathbf{x}_t) \right] \mathrm{d}t + g(t)\,\mathrm{d}\bar{W}_t, \tag{2}$$

where $\nabla_{\mathbf{x}_t} \log p_t(\mathbf{x}_t)$ is the score function. A neural network $s_\theta(\mathbf{x}_t, t)$ approximates the score via denoising score matching (Vincent, 2011):

$$s_\theta(\mathbf{x}_t, t) = -\frac{\epsilon_\theta(\mathbf{x}_t, t)}{\sigma_t} \approx \nabla_{\mathbf{x}_t} \log p_t(\mathbf{x}_t), \tag{3}$$

where $\epsilon_\theta$ predicts the added noise. Here we adopt the variance-preserving (VP) formulation $\mathbf{x}_t = \alpha_t \mathbf{x}_0 + \sigma_t \boldsymbol{\epsilon}$ with $\boldsymbol{\epsilon} \sim \mathcal{N}(\mathbf{0}, \mathbf{I})$, where $\alpha_t$ and $\sigma_t$ are schedule coefficients satisfying $\alpha_t^2 + \sigma_t^2 = 1$.

**Flow Matching.** Flow matching (Lipman et al., 2022) learns a continuous-time normalizing flow between a source distribution $\mu_0$ (typically Gaussian noise) and a target distribution $\mu_1$ (data) over $t \in [0, 1]$. Given samples $\mathbf{x}_0 \sim \mu_0$ and $\mathbf{x}_1 \sim \mu_1$, the conditional flow path follows the optimal transport interpolation $\mathbf{x}_t = (1 - t)\mathbf{x}_0 + t\mathbf{x}_1$, governed by an ODE:

$$\frac{\mathrm{d}\mathbf{x}_t}{\mathrm{d}t} = v_t(\mathbf{x}_t). \tag{4}$$

The model learns a velocity field $v_\theta$ by minimizing:

$$\mathcal{L}_\theta = \mathbb{E}_{t, \mathbf{x}_0, \mathbf{x}_1} \left[ \|v_\theta(\mathbf{x}_t, t) - (\mathbf{x}_1 - \mathbf{x}_0)\|^2 \right]. \tag{5}$$

During inference, samples are generated by integrating the ODE from $t = 0$ to $t = 1$, providing a deterministic alternative to the stochastic sampling in diffusion models.

### 3.2. Overview of EnsembleVLA

EnsembleVLA is an energy-based ensemble framework that combines diverse VLA policies into a unified, stronger policy. As illustrated in Figure 2, our approach consists of three components: (1) a *unified energy-based framework* that formulates both diffusion and flow-based models under a common formalism; (2) a *compositional generation* method that aggregates policies at the distribution level through additive energy combination; and (3) a *learnable composition weights and a confidence-aware gating mechanism* for residual correction. We describe each element of the framework in detail below.

### 3.3. Unified Energy-Based Theoretical Framework

To enable principled composition of diverse VLA models, we first establish a unified theoretical framework showing that both diffusion-based and flow-based VLA models can be interpreted as energy-based models, where the score function and velocity field both correspond to the negative energy gradient.

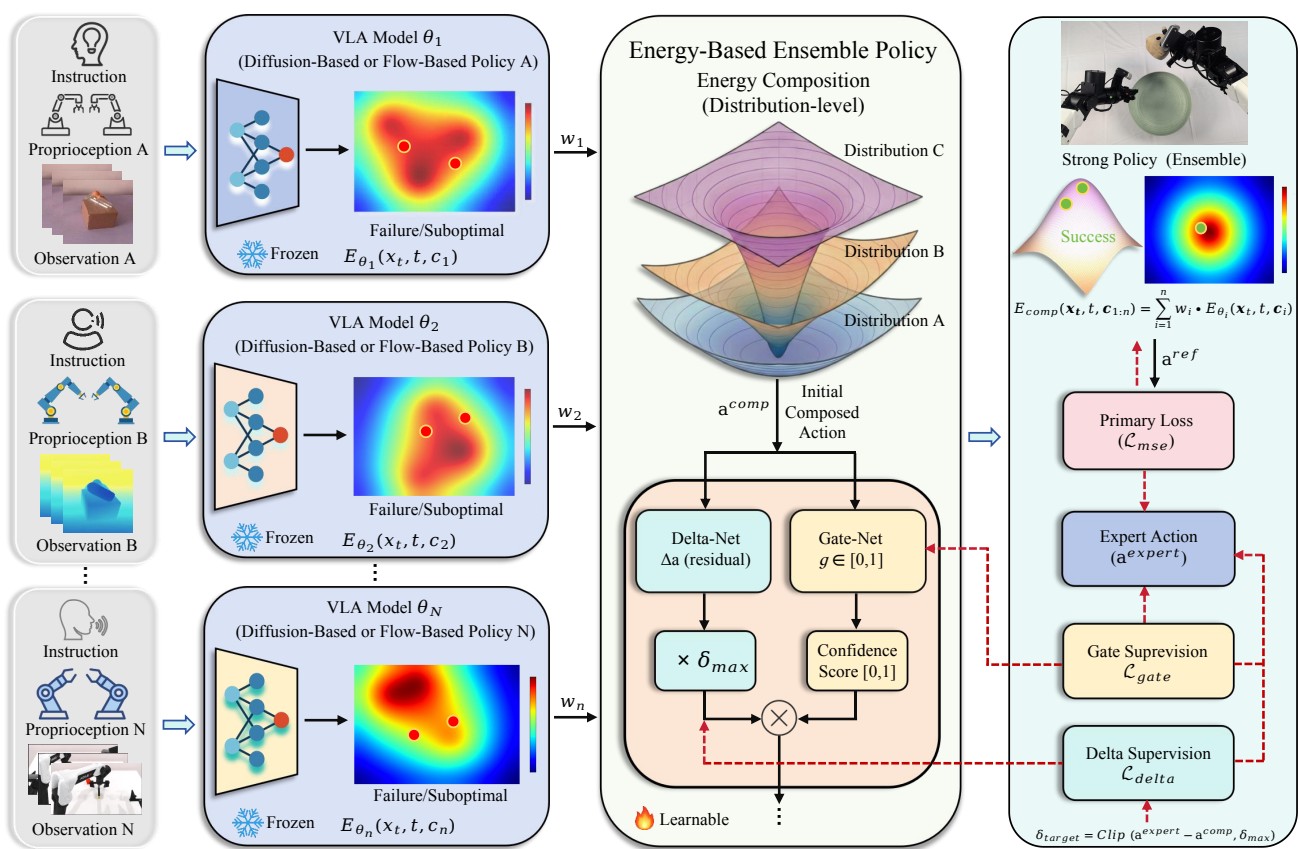

*Figure 2.* **Overall of EnsembleVLA**. We model diverse VLA models as independent energy functions and compose them at the distribution level to form a unified ensemble policy. To refine the composed actions, we introduce a confidence-aware gating mechanism with a Delta-Net predicting bounded residual corrections and a Gate-Net estimating refinement confidence.

**Energy-Based Model Formulation.** Energy-based models (EBMs) (Du & Mordatch, 2019) provide a unified perspective for understanding both diffusion-based and flow-based policies. An EBM defines a probability distribution via a learnable energy function. For a static distribution, the energy function $E_\theta(\mathbf{x})$ defines:

$$p_\theta(\mathbf{x}) = \frac{\exp\big(-E_\theta(\mathbf{x})\big)}{Z_\theta}, \qquad (6)$$

where $Z_\theta = \int \exp(-E_\theta(\mathbf{x}))\,\mathrm{d}\mathbf{x}$ is the partition function. In the context of generative models, we extend this to time-dependent energy functions $E_\theta(\mathbf{x}_t, t)$ capturing the evolving distribution during generation:

$$p_t(\mathbf{x}_t) \propto \exp\big(-E_\theta(\mathbf{x}_t, t)\big). \qquad (7)$$

A fundamental property of EBMs is that the score function equals the negative energy gradient:

$$\nabla_{\mathbf{x}_t} \log p_t(\mathbf{x}_t) = -\nabla_{\mathbf{x}_t} E_\theta(\mathbf{x}_t, t). \qquad (8)$$

This identity allows us to interpret score-based generative models as performing gradient descent on an implicit energy landscape.

**Diffusion Models as Energy-Based Models.** We interpret diffusion models within the energy-based framework by representing the energy function through the denoising network. In diffusion models, the reverse process aims to denoise $\mathbf{x}_t$ to $\mathbf{x}_{t-\Delta t}$ by predicting the noise $\epsilon_\theta(\mathbf{x}_t, t)$. By leveraging the Gaussian denoising formulation, the denoising step can be written as:

$$\mathbf{x}_{t-\Delta t} = \mathbf{x}_t - \eta_t \cdot \frac{\epsilon_\theta(\mathbf{x}_t, t)}{\sigma_t} + \sqrt{2\eta_t}\,\epsilon, \qquad (9)$$

where $\epsilon \sim \mathcal{N}(0, \mathbf{I})$, and $\eta_t > 0$ is a time-dependent step size. This corresponds to Langevin dynamics (Du & Mordatch, 2019):

$$\mathbf{x}_{t-\Delta t} = \mathbf{x}_t - \eta_t \nabla_{\mathbf{x}_t} E_\theta(\mathbf{x}_t, t) + \sqrt{2\eta_t}\,\epsilon, \qquad (10)$$

with the energy gradient given by $\nabla_{\mathbf{x}_t} E_\theta = \epsilon_\theta / \sigma_t$. In the deterministic limit (when the noise term vanishes), the relationship becomes:

$$s_\theta(\mathbf{x}_t, t) = -\frac{\epsilon_\theta(\mathbf{x}_t, t)}{\sigma_t} = -\nabla_{\mathbf{x}_t} E_\theta(\mathbf{x}_t, t), \qquad (11)$$

establishing that the score function directly corresponds to the negative energy gradient.

**Flow Matching as Energy-Based Models.** We establish the connection between flow matching and energy-based formulations through the Helmholtz decomposition. Any smooth velocity field can be decomposed as:

$$v_\theta(\mathbf{x}_t, t) = -\nabla_{\mathbf{x}_t} \Phi_\theta(\mathbf{x}_t, t) + \mathbf{r}(\mathbf{x}_t, t), \qquad (12)$$

where $\Phi_\theta$ is a scalar potential and $\mathbf{r}$ is a divergence-free residual. By identifying $E_\theta := \Phi_\theta$, we obtain the energy-based interpretation:

$$v_\theta(\mathbf{x}_t, t) = -\nabla_{\mathbf{x}_t} E_\theta(\mathbf{x}_t, t) + \mathbf{r}(\mathbf{x}_t, t). \qquad (13)$$

Under optimal transport objectives, Brenier's theorem (Brenier, 1991) guarantees that the optimal velocity field is conservative ($\mathbf{r} = \mathbf{0}$). While neural approximations yield non-zero $\mathbf{r}$, empirical studies (Balcerak et al., 2025) show $\|\mathbf{r}\| \ll \|v_\theta\|$, validating the approximation $v_\theta \approx -\nabla E_\theta$ in practice.

**Unified Framework.** Combining Eq. (8) and Eq. (13), we arrive at a unified energy-based perspective:

$$
\begin{aligned}
s_\theta(\mathbf{x}_t, t) &= -\nabla_{\mathbf{x}_t} E_\theta(\mathbf{x}_t, t), \\
v_\theta(\mathbf{x}_t, t) &= -\nabla_{\mathbf{x}_t} E_\theta(\mathbf{x}_t, t) + \mathbf{r}(\mathbf{x}_t, t),
\end{aligned}
\qquad (14)
$$

This unified perspective reveals that both paradigms perform gradient descent on an energy landscape—diffusion via stochastic Langevin dynamics and flow matching via deterministic gradient flow. Although they operate in opposite temporal directions (diffusion: $T \to 0$; flow matching: $0 \to 1$), both can be unified through time reparameterization, enabling principled composition of both paradigms.

### 3.4. Compositional Generation

Building upon the unified energy-based framework, we now present how EnsembleVLA composes multiple VLA policies through additive energy combination. Given $n$ policies characterized by conditional energy functions $\{E_{\theta_i}(\mathbf{x}_t, t, \mathbf{c}_i)\}_{i=1}^n$, where $\mathbf{c}_i$ denotes the conditioning information (e.g., visual observations, language instructions) for the $i$-th policy, the compositional energy is defined as:

$$E_{\text{comp}}(\mathbf{x}_t, t, \mathbf{c}_{1:n}) = \sum_{i=1}^n w_i E_{\theta_i}(\mathbf{x}_t, t, \mathbf{c}_i), \qquad (15)$$

where $w_i$ are learnable composition weights that will be optimized during training to enable adaptive policy balancing (detailed in Section 3.5), and $\mathbf{c}_{1:n} = (\mathbf{c}_1, \dots, \mathbf{c}_n)$ denotes the collection of conditioning information.

Drawing from the compositional generation principle in EBMs (Du et al., 2020; Zhang et al., 2025a), the corresponding conditional probability distribution becomes:

$$p(\mathbf{x}_t \mid \mathbf{c}_{1:n}) \propto \prod_{i=1}^n p_{\theta_i}(\mathbf{x}_t \mid \mathbf{c}_i)^{w_i}, \qquad (16)$$

representing a geometric mixture when $\sum_i w_i = 1$ (as enforced by our softmax parameterization in Section 3.5). Taking the gradient with respect to $\mathbf{x}_t$ yields the compositional score field:

$$
\begin{aligned}
s_{\text{comp}}(\mathbf{x}_t, t, \mathbf{c}_{1:n}) &= \sum_{i=1}^n w_i \cdot \nabla_{\mathbf{x}_t} \log p_{\theta_i}(\mathbf{x}_t \mid \mathbf{c}_i) \\
&= \sum_{i=1}^n w_i \cdot s_{\theta_i}(\mathbf{x}_t, t, \mathbf{c}_i).
\end{aligned}
\qquad (17)
$$

By the unified energy-based framework (Eq. (14)), the same linear combination principle applies to velocity fields: $v_{\text{comp}} = \sum_{i=1}^n w_i \cdot v_{\theta_i}$. We sample the composed action $\mathbf{a}^{\text{comp}}$ via iterative integration from $\mathbf{x}_0 \sim \mathcal{N}(0, \mathbf{I})$ using a unified progress variable $\tau \in [0, 1]$:

$$\mathbf{x}_{\tau + \Delta\tau} = \mathbf{x}_\tau + \Delta\tau \cdot v_{\text{comp}}(\mathbf{x}_\tau, \tau, \mathbf{c}_{1:n}), \qquad (18)$$

which corresponds to solving the probability flow ODE. We use deterministic sampling for all compositions as it provides more stable action generation for robotic control. The final sample $\mathbf{x}_1$ at $\tau = 1$ yields the composed action $\mathbf{a}^{\text{comp}} := \mathbf{x}_1$. For combining different model types, we use displacement-space composition to align predictions before composition.

### 3.5. Ensemble Refinement

The composed action $\mathbf{a}^{\text{comp}} \in \mathbb{R}^{D_a}$ ($D_a$ is the action dimension) may deviate from optimal due to distribution mismatch or complementary errors. To address this, EnsembleVLA incorporates two refinement mechanisms: (1) *learnable composition weights* for adaptive policy balancing, and (2) a *confidence-aware gating mechanism* that selectively activates bounded residual corrections. Let $\phi = (\phi_\delta, \phi_g)$ denote the refinement module parameters. The refined action is:

$$\mathbf{a}^{\text{ref}} = \mathbf{a}^{\text{comp}} + g_{\phi_g}(\mathbf{a}^{\text{comp}}) \cdot \boldsymbol{\delta}_{\phi_\delta}(\mathbf{a}^{\text{comp}}), \qquad (19)$$

where $g_{\phi_g} : \mathbb{R}^{D_a} \to [0, 1]$ denotes the confidence-aware gating function, and $\boldsymbol{\delta}_{\phi_\delta} : \mathbb{R}^{D_a} \to [-\delta_{\max}, \delta_{\max}]^{D_a}$ represents the bounded residual correction function.

**Learnable Composition Weights for Adaptive Policy Balancing.** For adaptive policy balancing, we introduce learnable weights $\mathbf{w} = (w_1, \dots, w_n)^\top$ optimized end-to-end.

To ensure the weights reside on the probability simplex $\mathcal{S}^{n-1} = \{\mathbf{w} \in \mathbb{R}_+^n : \mathbf{1}^\top \mathbf{w} = 1\}$, we employ the softmax reparameterization:

$$w_i = \frac{\exp(\alpha_i)}{\sum_{j=1}^n \exp(\alpha_j)}, \quad i = 1, \dots, n, \qquad (20)$$

where $\boldsymbol{\alpha} \in \mathbb{R}^n$ denotes unconstrained learnable parameters. This ensures normalized weights that maintain contributions from all policies and yield stable sampling. The weights $\{w_i\}_{i=1}^n$ are jointly optimized with the refinement networks.

**Bounded Residual Correction.** The residual correction function $\boldsymbol{\delta}_{\phi_\delta}$ predicts action corrections that compensate for the residual error between the composed action and the optimal action. To prevent over-correction and ensure training stability, we constrain the correction to a bounded output space $\mathcal{H}_\delta = \{\boldsymbol{\delta} : \|\boldsymbol{\delta}\|_\infty \leq \delta_{\max}\}$ via:

$$\boldsymbol{\delta}_{\phi_\delta}(\mathbf{a}) = \delta_{\max} \cdot \tanh\big(f_\delta(\mathbf{a}; \phi_\delta)\big), \qquad (21)$$

where the Delta Network $f_\delta : \mathbb{R}^{D_a} \to \mathbb{R}^{D_a}$ is parameterized as a two-layer MLP with Spectral Normalization (Miyato et al., 2018) to enforce Lipschitz continuity (Hager, 1979), ensuring that small perturbations in the input induce only bounded variations in the output. We set $\delta_{\max} = 0.001$ for conservative corrections.

**Confidence-Aware Gating Mechanism.** The Gate Network $g_{\phi_g}$ predicts a scalar confidence score that modulates whether to apply the residual correction:

$$g_{\phi_g}(\mathbf{a}) = \sigma\big(f_g(\mathbf{a}; \phi_g)\big), \qquad (22)$$

where $\sigma(\cdot)$ denotes the sigmoid function and $f_g : \mathbb{R}^{D_a} \to \mathbb{R}$ is implemented as a three-layer MLP. Intuitively, the gating mechanism learns to assess when correction is beneficial: when $\mathbf{a}^{\mathrm{comp}}$ already aligns well with the optimal action, the gate outputs a low value to preserve it; conversely, when $\mathbf{a}^{\mathrm{comp}}$ deviates significantly, a high gate value activates the residual correction. This selective activation prevents unnecessary modifications that could degrade performance. We initialize the bias to $b_0 = -2.0$ (yielding $g_0 = \sigma(-2.0) \approx 0.12$) for conservative startup.

### 3.6. Training Objective

To train EnsembleVLA, we design a composite loss function consisting of three complementary terms:

$$\mathcal{L} = \mathcal{L}_{\mathrm{mse}} + \lambda_1 \mathcal{L}_{\mathrm{gate}} + \lambda_2 \mathcal{L}_{\mathrm{delta}}, \qquad (23)$$

where $\lambda_1$ and $\lambda_2$ are balancing coefficients that control the relative importance of the gating and residual prediction objectives, respectively.

**Primary Reconstruction Loss.** The primary loss measures the mean squared error between the refined action and the expert demonstration:

$$\mathcal{L}_{\mathrm{mse}} = \frac{1}{B} \sum_{i=1}^{B} \big\|\mathbf{a}_i^{\mathrm{ref}} - \mathbf{a}_i^{\mathrm{expert}}\big\|^2, \qquad (24)$$

where $B$ denotes the batch size. This loss drives the network to learn effective correction strategies.

**Gate Supervision Loss.** To teach the Gate Network when to apply corrections, we introduce a gate supervision loss based on predicted improvement estimation. We compute the improvement metric:

$$\Delta_i = \big\|\mathbf{a}_i^{\mathrm{comp}} - \mathbf{a}_i^{\mathrm{expert}}\big\|^2 - \big\|\mathbf{a}_i^{\mathrm{comp}} + \boldsymbol{\delta}_i - \mathbf{a}_i^{\mathrm{expert}}\big\|^2, \qquad (25)$$

where $\boldsymbol{\delta}_i = \boldsymbol{\delta}_{\phi_\delta}(\mathbf{a}_i^{\mathrm{comp}})$ is the predicted residual correction. A positive $\Delta_i$ indicates that applying the correction would bring the action closer to the expert demonstration. The binary gate supervision target is:

$$y_i^{\mathrm{gate}} = \mathbb{1}[\Delta_i > 0], \qquad (26)$$

where we detach the gradient through $\boldsymbol{\delta}_i$ during backpropagation. The gate loss is formulated using binary cross-entropy:

$$\mathcal{L}_{\mathrm{gate}} = -\frac{1}{B} \sum_{i=1}^{B} \big[ y_i^{\mathrm{gate}} \log(g_i) + (1 - y_i^{\mathrm{gate}}) \log(1 - g_i) \big], \qquad (27)$$

where $g_i = g_{\phi_g}(\mathbf{a}_i^{\mathrm{comp}})$ and $y_i^{\mathrm{gate}} \in \{0, 1\}$ indicates which paradigm yields lower action error for sample $i$.

**Delta Supervision Loss.** To accelerate convergence, we directly supervise the Delta Network with the bounded optimal correction. The target correction is:

$$\boldsymbol{\delta}_i^{\mathrm{target}} = \mathrm{clip}\big(\mathbf{a}_i^{\mathrm{expert}} - \mathbf{a}_i^{\mathrm{comp}}, -\delta_{\max}, \delta_{\max}\big), \qquad (28)$$

and we minimize the mean squared error:

$$\mathcal{L}_{\mathrm{delta}} = \frac{1}{B} \sum_{i=1}^{B} \big\|\boldsymbol{\delta}_i - \boldsymbol{\delta}_i^{\mathrm{target}}\big\|^2. \qquad (29)$$

This loss provides a direct learning signal for the Delta-Net, complementing the end-to-end supervision from $\mathcal{L}_{\mathrm{mse}}$.

## 4. Experiments

### 4.1. Datasets and Experimental Setup

We evaluate EnsembleVLA on eight simulated tasks in RoboTwin 2.0 (Chen et al., 2025) and six real-world tasks on the Cobot Mobile ALOHA (Fu et al., 2024). The simulated tasks cover a broad spectrum of bimanual coordination challenges, including coordinated grasping, sequential object transfer, and articulated object interaction. We adopt the standard benchmark protocol: training is performed with 100 expert demonstrations per task, and evaluation is conducted over 100 episodes per task in simulation and 20 trials per task in real-world settings.

**Baselines.** We benchmark EnsembleVLA against four representative VLA models that span different generative

*Table 1.* **Experiment results on different policy compositions.** The table shows the success rate ↑. Our EnsembleVLA yields a noticeable improvement compared with the base policies. We report the average success rate and sample standard deviation over 3 seeds.

| Task | DP + DP3 (Diffusion-Based + Diffusion-Based) | | | DP + $\pi$0.5 (Diffusion-Based + Flow-Based) | | | $\pi$0 + $\pi$0.5 (Flow-Based + Flow-Based) | | |
|---|---|---|---|---|---|---|---|---|---|
| | DP | DP3 | **EnsembleVLA** | DP | $\pi$0.5 | **EnsembleVLA** | $\pi$0 | $\pi$0.5 | **EnsembleVLA** |
| Beat Block Hammer | 64.0±2.3 | 85.0±1.7 | **97.0**±1.0 | 64.0±2.3 | 67.0±2.0 | **73.0**±1.5 | 62.0±1.2 | 67.0±2.0 | **73.0**±1.3 |
| Open Laptop | 67.0±2.1 | 82.0±1.5 | **93.0**±1.2 | 67.0±2.1 | 93.0±1.0 | **95.0**±0.9 | 85.0±1.8 | 93.0±1.0 | **97.0**±1.4 |
| Click Alarmclock | 92.0±1.3 | 76.0±2.2 | **96.0**±1.6 | 92.0±1.3 | 84.0±1.8 | **94.0**±1.1 | 52.0±2.5 | 84.0±1.8 | **91.0**±1.7 |
| Move Playingcard Away | 61.0±2.7 | 64.0±2.4 | **89.0**±1.9 | 61.0±2.7 | 82.0±1.6 | **88.0**±1.4 | 56.0±1.5 | 82.0±1.6 | **89.0**±1.8 |
| Place Bread Skillet | 27.0±1.9 | 46.0±2.6 | **55.0**±2.2 | 27.0±1.9 | 50.0±2.3 | **52.0**±1.6 | 30.0±2.1 | 50.0±2.3 | **61.0**±2.4 |
| Dump Bin Bigbin | 53.0±2.8 | 78.0±1.9 | **88.0**±1.2 | 53.0±2.8 | 69.0±2.0 | **72.0**±1.3 | 65.0±1.6 | 69.0±2.0 | **75.0**±2.0 |
| Handover Block | 17.0±1.2 | 49.0±2.7 | **70.0**±2.5 | 17.0±1.2 | 31.0±2.1 | **40.0**±2.3 | 28.0±1.9 | 31.0±2.1 | **54.0**±2.6 |
| Stack Bowls Three | 35.0±2.6 | 67.0±2.1 | **74.0**±1.5 | 35.0±2.6 | 45.0±2.4 | **47.0**±1.7 | 42.0±1.8 | 45.0±2.4 | **57.0**±2.3 |
| **Average** | 52.0±2.1 | 68.4±2.1 | **82.8**±1.6 | 52.0±2.1 | 65.1±1.9 | **70.1**±1.5 | 52.5±1.8 | 65.1±1.9 | **74.6**±1.9 |
| **Improvement** | | +14.4 | | | +5.0 | | | +9.5 | |

*Table 2.* Ablation study of EnsembleVLA components on DP + DP3 ensemble. We report success rate (%) over 3 seeds.

| Compose | Weights | Delta | Gate | Success Rate (%) ↑ | | |
|---|---|---|---|---|---|---|
| | | | | Beat Block Hammer | Handover Block | Place Bread Skillet |
| ✓ | ✓ | ✓ | ✓ | **97.0**±1.0 | **70.0**±2.5 | **55.0**±2.2 |
| ✓ | ✓ | ✓ | ✗ | 93.0±2.0 | 64.0±3.5 | 50.0±4.0 |
| ✓ | ✓ | ✗ | ✗ | 89.0±1.3 | 56.0±1.0 | 43.0±1.6 |
| ✓ | ✗ | ✗ | ✗ | 86.0±1.4 | 42.0±2.2 | 38.0±4.5 |
| ✗ | ✗ | ✗ | ✗ | 26.0±3.0 | 12.0±1.8 | 14.0±1.5 |

paradigms. Specifically, we consider: (1) DP (Chi et al., 2025), which formulates visuomotor control as a conditional denoising process over the action space; (2) DP3 (Ze et al., 2024), an extension that leverages point cloud inputs for explicit geometric reasoning; and (3) $\pi_0$ (Black et al., 2024) and $\pi_{0.5}$ (Intelligence et al., 2025), two recent flow matching-based foundation models trained on diverse robotic manipulation data. To ensure fair evaluation, all policies are trained on identical demonstration data. Pre-trained models ($\pi_0$ and $\pi_{0.5}$) are initialized from publicly available weights and subsequently fine-tuned, whereas DP and DP3 are trained from random initialization.

**Results.** Quantitative results are summarized in Table 1. We evaluate EnsembleVLA under three distinct policy compositions: (1) two diffusion-based policies (DP + DP3), (2) a diffusion-based and a flow-based policy (DP + $\pi_{0.5}$), and (3) two flow-based policies ($\pi_0$ + $\pi_{0.5}$). EnsembleVLA consistently outperforms all base policies across all compositions. In the diffusion-based combination, EnsembleVLA achieves 82.8%, improving 14.4% over DP3 (68.4%). When combining two flow-based policies, EnsembleVLA attains 74.6%, surpassing $\pi_{0.5}$ by 9.5%. These gains demonstrate that our energy-based composition effectively aggregates complementary information from policies within the same

generative family. For the cross-paradigm composition, EnsembleVLA achieves 70.1%, a 5.0% improvement over $\pi_{0.5}$. This relatively modest gain is attributed to the additional complexity of velocity field transformation when composing diffusion-based and flow-based policies. Nevertheless, the positive results confirm that EnsembleVLA can successfully bridge different generative paradigms, validating its effectiveness in ensembling diverse VLA models to produce a stronger unified policy.

### 4.2. Ablation Study

To evaluate the contribution of each component, we conduct ablation experiments on three representative tasks. We progressively remove components to analyze their individual contributions. As shown in Table 2, the complete model achieves the best performance across all tasks. Removing the Gate mechanism leads to moderate degradation (70.0% → 64.0% on Handover Block), as the model loses selective correction capability and applies modifications indiscriminately. Further removing Delta causes additional performance drops, demonstrating that bounded residual corrections are essential for compensating errors in the composed actions. When Learnable Weights are also removed, performance degrades substantially on tasks where base policies differ significantly in quality. For example, on Handover Block where DP achieves only 17.0%, fixed equal weighting fails to suppress the weak policy's contribution, resulting in a 14% drop.

Most notably, replacing energy-based composition with naive action-level averaging causes dramatic performance collapse (e.g., 42.0% → 12.0% on Handover Block). This is because directly averaging actions from different policies produces geometrically meaningless outputs that fail to represent valid robot motions. These results validate that each component contributes meaningfully, and their combination is essential for effective policy ensemble.

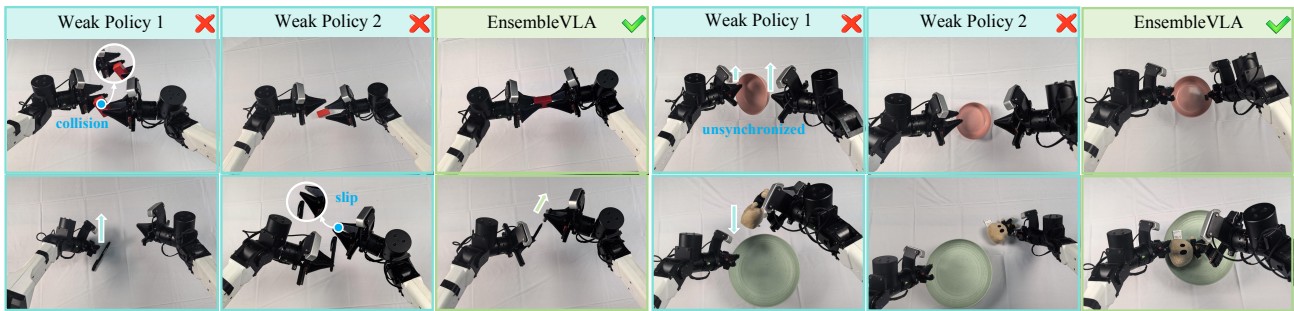

*Figure 3.* Real-world bimanual manipulation: individual weak policies vs. EnsembleVLA. EnsembleVLA successfully handles tasks where individual policies fail (collision, slip, unsynchronized).

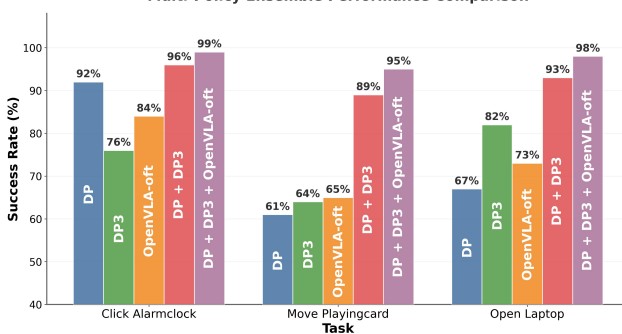

*Figure 4.* **Multi-policy ensemble performance.** Success rates of individual policies and their ensembles on three manipulation tasks. Combining more policies consistently improves performance.

### 4.3. Scaling Up to Multi-Policy Ensembles

To validate the scalability of our framework, we extend the composition to three VLA models: DP, DP3, and OpenVLA-OFT. We evaluate on three manipulation tasks: *click alarmclock*, *move playingcard*, and *open laptop*. As shown in Figure 4, individual policies achieve success rates ranging from 61% to 92% across different tasks. When composing DP and DP3 through our unified energy formulation, the ensemble achieves 96%, 89%, and 93% on the three tasks, consistently outperforming the individual baselines. Furthermore, incorporating OpenVLA-OFT into the ensemble yields additional improvements, reaching 99%, 95%, and 98%, respectively. These results demonstrate that our framework effectively leverages the complementary strengths of diverse policies, and the performance gains scale favorably with the number of composed models.

### 4.4. Real-World Experiments

To validate the effectiveness of EnsembleVLA in real-world scenarios, we conduct experiments on a Cobot Mobile ALOHA across six manipulation tasks: *handover a cube*, *open the pen cap*, *pick up bowl*, *push box*, *put the banana into the drawer*, and *put the doll into the plate*. We collect

*Table 3.* Real-world manipulation tasks performance comparison. We report success rates (%) over 20 trials per task.

| Method | Handover Cube | Open Pen Cap | Pick up Bowl | Push Box | Put Banana into Drawer | Put Doll into Plate | Avg. |
|---|---|---|---|---|---|---|---|
| DP | 45.0 | 30.0 | 50.0 | 55.0 | 25.0 | 35.0 | 40.0 |
| $\pi_{0.5}$ | 60.0 | 50.0 | 65.0 | 70.0 | 45.0 | 55.0 | 57.5 |
| EnsembleVLA | **70.0** | **60.0** | **75.0** | **80.0** | **50.0** | **60.0** | **65.8** |

50 demonstrations per task for training and evaluate each method over 20 trials per task. As shown in Table 3, EnsembleVLA consistently outperforms individual policies (DP and $\pi_{0.5}$) across all tasks. Figure 3 provides qualitative comparisons, where individual policies exhibit various failure modes including collisions, object slipping, and unsynchronized bimanual coordination. By composing these policies through our energy-based framework, EnsembleVLA successfully overcomes these failure modes and achieves robust performance, demonstrating the effectiveness of our compositional approach in leveraging complementary strengths of different policies.

## 5. Conclusion

This paper presents EnsembleVLA, a novel framework that combines diverse vision-language-action policies through energy-based composition. Our core contribution is a theoretical unification of diffusion and flow matching VLA paradigms, demonstrating that both can be formulated as energy-based models. This insight enables principled policy aggregation via weighted energy summation at the distribution level. We further introduce adaptive weighting and gating mechanisms to refine composed actions through dynamic policy balancing. Experimental results on simulation benchmarks and real-world robot deployments show that EnsembleVLA consistently outperforms individual base policies, validating the effectiveness of our approach for robotic manipulation. We hope EnsembleVLA inspires more researchers to explore the potential of ensemble learning in the field of embodied AI.

## Acknowledgements

We would like to thank all co-authors for their efforts and the reviewers for their constructive comments. This work is supported by the National Natural Science Foundation of China (Key Project of Joint Fund) (Grant No. U24A20328), the National Natural Science Foundation of China (Grant No. 62476071), the Guangdong Basic and Applied Basic Research Foundation (Grant No. 2025A1515011732), the Beijing Natural Science Foundation (Grant Nos. 4262074 and L254018), the National Natural Science Foundation of China (Grant No. 62406092), the National Natural Science Foundation of China (Grant No. U24B20175), the Shenzhen Science and Technology Program (Grant No. KJZD20240903100017022), the Guangdong Basic and Applied Basic Research Foundation (Grant No. 2025A1515010169), and the Shenzhen Science and Technology Program (Grant No. KQTD20240729102207002).

## Impact Statement

This paper presents EnsembleVLA, an energy-based framework for combining diverse vision-language-action models to enhance robotic manipulation capabilities. By enabling the effective integration of multiple pre-trained VLA policies, our approach has the potential to accelerate the deployment of general-purpose robots in real-world applications such as manufacturing automation, logistics, healthcare assistance, and domestic service robotics, where robust manipulation capabilities are essential. The ability to combine complementary strengths of different policies may also reduce the data and computational requirements for training new robotic systems from scratch. As with any advancement in robotic manipulation, there are considerations regarding workforce displacement in automation-heavy industries and the safety of human-robot collaboration scenarios.

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

In this Supplementary Material, we provide detailed derivations of the ensemble energy-based framework (Section A), pseudocode for our unified cross-paradigm (diffusion-based and flow-based) policy composition algorithm (Section B), extensive ablation studies on inference latency, loss weighting, residual bounds, and composition dynamics (Section C), a comprehensive description of the simulated bimanual manipulation tasks (Section D), details of the real-world hardware setup and evaluation tasks (Section E), and a discussion of current limitations along with future research directions (Sections F and G).

## A. Detailed Derivations

### A.1. Reverse-Time SDE

Given the forward SDE:

$$d\mathbf{x}_t = f(\mathbf{x}_t, t)\, dt + g(t)\, dW_t, \tag{A.1}$$

where $f(\mathbf{x}_t, t)$ is the drift, $g(t)$ is the diffusion coefficient, and $W_t$ is a Wiener process. By Anderson's theorem (Anderson, 1982), the reverse-time SDE is:

$$d\mathbf{x}_t = \left[ f(\mathbf{x}_t, t) - g(t)^2 \nabla_{\mathbf{x}_t} \log p_t(\mathbf{x}_t) \right] dt + g(t)\, d\bar{W}_t. \tag{A.2}$$

### A.2. Score Function and Noise Prediction

For the variance-preserving (VP) forward process:

$$\mathbf{x}_t = \alpha_t \mathbf{x}_0 + \sigma_t \boldsymbol{\epsilon}, \quad \boldsymbol{\epsilon} \sim \mathcal{N}(\mathbf{0}, \mathbf{I}), \tag{A.3}$$

where $\alpha_t^2 + \sigma_t^2 = 1$. The conditional distribution is:

$$p_t(\mathbf{x}_t \mid \mathbf{x}_0) = \mathcal{N}(\mathbf{x}_t; \alpha_t \mathbf{x}_0, \sigma_t^2 \mathbf{I}). \tag{A.4}$$

The conditional score is:

$$\nabla_{\mathbf{x}_t} \log p_t(\mathbf{x}_t \mid \mathbf{x}_0) = -\frac{\mathbf{x}_t - \alpha_t \mathbf{x}_0}{\sigma_t^2} = -\frac{\boldsymbol{\epsilon}}{\sigma_t}. \tag{A.5}$$

By denoising score matching (Vincent, 2011), the neural network $\epsilon_\theta(\mathbf{x}_t, t)$ predicts $\boldsymbol{\epsilon}$. We parameterize the score network via an energy function $E_\theta(\mathbf{x}_t, t)$:

$$s_\theta(\mathbf{x}_t, t) = -\nabla_{\mathbf{x}_t} E_\theta(\mathbf{x}_t, t) = -\frac{\epsilon_\theta(\mathbf{x}_t, t)}{\sigma_t}, \tag{A.6}$$

which is trained to approximate the true score $\nabla_{\mathbf{x}_t} \log p_t(\mathbf{x}_t)$.

### A.3. Probability Flow ODE

Song et al. (Song et al., 2020b) showed that the following ODE shares the same marginals with Eq. (A.2):

$$\frac{d\mathbf{x}_t}{dt} = f(\mathbf{x}_t, t) - \frac{1}{2} g(t)^2 \nabla_{\mathbf{x}_t} \log p_t(\mathbf{x}_t). \tag{A.7}$$

### A.4. Score Function and Energy Gradient

An EBM defines the distribution via energy function $E_\theta(\mathbf{x}_t, t)$:

$$p_t(\mathbf{x}_t) = \frac{\exp\left(-E_\theta(\mathbf{x}_t, t)\right)}{Z_\theta(t)}, \tag{A.8}$$

where $Z_\theta(t) = \int \exp(-E_\theta(\mathbf{x}, t))\, d\mathbf{x}$. Taking logarithm:

$$\log p_t(\mathbf{x}_t) = -E_\theta(\mathbf{x}_t, t) - \log Z_\theta(t). \tag{A.9}$$

Since $Z_\theta(t)$ is independent of $\mathbf{x}_t$, the gradient yields:

$$\nabla_{\mathbf{x}_t} \log p_t(\mathbf{x}_t) = -\nabla_{\mathbf{x}_t} E_\theta(\mathbf{x}_t, t). \tag{A.10}$$

## A.5. Velocity Field in Energy-Based Form

We establish the energy-based interpretation of flow matching through two complementary perspectives.

**Perspective 1: Helmholtz Decomposition.** By the Helmholtz decomposition theorem, any sufficiently smooth vector field $v_\theta : \mathbb{R}^d \to \mathbb{R}^d$ can be uniquely decomposed as:

$$v_\theta(\mathbf{x}_t, t) = -\nabla_{\mathbf{x}_t} \Phi(\mathbf{x}_t, t) + \nabla \times \mathbf{A}(\mathbf{x}_t, t), \tag{A.11}$$

where $\Phi : \mathbb{R}^d \to \mathbb{R}$ is a scalar potential (the conservative/irrotational component) and $\mathbf{A} : \mathbb{R}^d \to \mathbb{R}^d$ is a vector potential (the solenoidal/rotational component). Defining $E_\theta := \Phi$ and $\mathbf{r} := \nabla \times \mathbf{A}$, we obtain:

$$v_\theta(\mathbf{x}_t, t) = -\nabla_{\mathbf{x}_t} E_\theta(\mathbf{x}_t, t) + \mathbf{r}(\mathbf{x}_t, t), \tag{A.12}$$

where $\nabla \cdot \mathbf{r} = 0$ (divergence-free) and $\nabla \times (-\nabla E_\theta) = \mathbf{0}$ (curl-free).

**Perspective 2: Optimal Transport Theory.** Under optimal transport with quadratic cost $c(\mathbf{x}, \mathbf{y}) = \|\mathbf{x} - \mathbf{y}\|^2 / 2$, Brenier's theorem (Brenier, 1991) states that the optimal transport map $T^*$ is the gradient of a convex function: $T^* = \nabla \Psi^*$ for some convex $\Psi^*$. The corresponding velocity field satisfies:

$$v^*(\mathbf{x}_t, t) = \frac{T^*(\mathbf{x}_0) - \mathbf{x}_0}{1} = \nabla \Psi^*(\mathbf{x}_0) - \mathbf{x}_0, \tag{A.13}$$

which is purely conservative ($\mathbf{r} = \mathbf{0}$). While neural network approximations $v_\theta$ do not exactly satisfy this condition, they inherit an inductive bias toward conservative fields when trained with OT-based objectives.

**Approximation Quality.** Combining both perspectives, we write:

$$v_\theta(\mathbf{x}_t, t) = -\nabla_{\mathbf{x}_t} E_\theta(\mathbf{x}_t, t) + \mathbf{r}(\mathbf{x}_t, t), \tag{A.14}$$

where the approximation $v_\theta \approx -\nabla E_\theta$ holds when $\|\mathbf{r}\| \ll \|v_\theta\|$. Empirical studies (Balcerak et al., 2025) confirm that flow matching with linear interpolation and Gaussian source distribution yields velocity fields with small rotational components, validating this approximation in practice.

## A.6. Temporal Unification

Diffusion generates from $t = T$ to $0$; flow matching from $t = 0$ to $1$. We unify via $\tau \in [0, 1]$:

$$t_{\text{diff}}(\tau) = T(1 - \tau), \quad t_{\text{flow}}(\tau) = \tau. \tag{A.15}$$

Both models then follow:

$$\frac{d\mathbf{x}_\tau}{d\tau} = \tilde{v}(\mathbf{x}_\tau, \tau). \tag{A.16}$$

## A.7. Compositional Score Function

Given $n$ policies with energy functions $\{E_{\theta_i}(\mathbf{x}_t, t, \mathbf{c}_i)\}_{i=1}^n$, the compositional energy is:

$$E_{\text{comp}}(\mathbf{x}_t, t, \mathbf{c}_{1:n}) = \sum_{i=1}^n w_i E_{\theta_i}(\mathbf{x}_t, t, \mathbf{c}_i). \tag{A.17}$$

The corresponding distribution:

$$p_{\text{comp}}(\mathbf{x}_t \mid \mathbf{c}_{1:n}) = \frac{\exp(-E_{\text{comp}}(\mathbf{x}_t, t, \mathbf{c}_{1:n}))}{Z_{\text{comp}}}. \tag{A.18}$$

---

**Algorithm 1** Cross-Paradigm Policy Composition

---

**Require:** Diffusion policy $\pi_{\mathrm{diff}}$, flow policy $\pi_{\mathrm{flow}}$, weights $(w_d, w_f)$, diffusion time $T$, steps $N$, noise schedule $\{\alpha_t, \sigma_t\}$, conditioning $\mathbf{c}$
**Ensure:** Composed action $\mathbf{a}^{\mathrm{comp}}$
1: $\mathbf{x}^{(0)} \sim \mathcal{N}(\mathbf{0}, \mathbf{I})$        $\triangleright$ Initialize from noise
2: **for** $i = 0$ to $N - 1$ **do**
3:     */\* Step 1: Temporal alignment \*/*
4:     $\tau \leftarrow i/N$        $\triangleright$ Unified progress
5:     $t_d \leftarrow T(1 - \tau), \quad t_d' \leftarrow T(1 - \tau - 1/N)$        $\triangleright$ Diffusion timestamps
6:     $t_f \leftarrow \tau$        $\triangleright$ Flow timestamp
7:     */\* Step 2: Model predictions \*/*
8:     $\boldsymbol{\epsilon} \leftarrow \pi_{\mathrm{diff}}(\mathbf{x}^{(i)}, t_d, \mathbf{c})$        $\triangleright$ Noise prediction
9:     $\mathbf{v} \leftarrow \pi_{\mathrm{flow}}(\mathbf{x}^{(i)}, t_f, \mathbf{c})$        $\triangleright$ Velocity prediction
10:     */\* Step 3: Convert to displacement space \*/*
11:     $\hat{\mathbf{x}}_0 \leftarrow (\mathbf{x}^{(i)} - \sigma_{t_d}\boldsymbol{\epsilon})/\alpha_{t_d}$        $\triangleright$ DDIM: estimate clean sample
12:     $\Delta\mathbf{x}_d \leftarrow \alpha_{t_d'}\hat{\mathbf{x}}_0 + \sigma_{t_d'}\boldsymbol{\epsilon} - \mathbf{x}^{(i)}$        $\triangleright$ Diffusion displacement
13:     $\Delta\mathbf{x}_f \leftarrow \mathbf{v}/N$        $\triangleright$ Flow displacement
14:     */\* Step 4: Weighted composition \*/*
15:     $\mathbf{x}^{(i+1)} \leftarrow \mathbf{x}^{(i)} + w_d \cdot \Delta\mathbf{x}_d + w_f \cdot \Delta\mathbf{x}_f$
16: **end for**
17: **return** $\mathbf{a}^{\mathrm{comp}} \leftarrow \mathbf{x}^{(N)}$

---

Expanding:

$$
p_{\mathrm{comp}}(\mathbf{x}_t \mid \mathbf{c}_{1:n}) \propto \exp\left(-\sum_{i=1}^{n} w_i E_{\theta_i}\right)
$$

$$
= \prod_{i=1}^{n} \exp(-w_i E_{\theta_i})
$$

$$
\propto \prod_{i=1}^{n} p_{\theta_i}(\mathbf{x}_t \mid \mathbf{c}_i)^{w_i}. \tag{A.19}
$$

Taking gradient:

$$
s_{\mathrm{comp}}(\mathbf{x}_t, t, \mathbf{c}_{1:n}) = \sum_{i=1}^{n} w_i \cdot s_{\theta_i}(\mathbf{x}_t, t, \mathbf{c}_i). \tag{A.20}
$$

By Eq. (A.10) and Eq. (A.14), the same applies to velocity fields:

$$
v_{\mathrm{comp}}(\mathbf{x}_t, t, \mathbf{c}_{1:n}) = \sum_{i=1}^{n} w_i \cdot v_{\theta_i}(\mathbf{x}_t, t, \mathbf{c}_i). \tag{A.21}
$$

## B. Composition of Diffusion-Based and Flow-Based VLA Models

To compose policies from different generative paradigms, we convert their outputs to a common displacement space. Diffusion models predict noise $\epsilon_\theta$, while flow matching models predict velocity $v_\theta$. Although these models are based on different formulations, with diffusion operating in a denoising framework and flow matching in a continuous trajectory setting, they share the same underlying goal: to guide the action generation process toward high-likelihood regions of the data distribution.

For diffusion, given noise prediction $\epsilon$ at time $t_d$, the DDIM update (Song et al., 2020a) estimates the clean sample and computes the displacement:

$$
\hat{\mathbf{x}}_0 = \frac{\mathbf{x}_t - \sigma_{t_d}\boldsymbol{\epsilon}}{\alpha_{t_d}}, \quad \Delta\mathbf{x}_d = \alpha_{t_d'}\hat{\mathbf{x}}_0 + \sigma_{t_d'}\boldsymbol{\epsilon} - \mathbf{x}_t. \tag{B.1}
$$

*Table 4.* **Inference latency and performance analysis on L40S GPU.** EnsembleVLA executes both policies in parallel with minimal additional overhead, while achieving substantial performance gains.

| Configuration | Latency | Relative | Frequency | Avg. Success |
|---|---|---|---|---|
| $\pi_0$ (single policy) | 100.0 ms | $1.00\times$ | 10.0 Hz | $52.5 \pm 1.8\%$ |
| $\pi_{0.5}$ (single policy) | 112.5 ms | $1.13\times$ | 8.9 Hz | $65.1 \pm 1.9\%$ |
| EnsembleVLA ($\pi_0 + \pi_{0.5}$) | 117.5 ms | $1.18\times$ | 8.5 Hz | $\mathbf{74.6 \pm 1.9\%}$ |

*Table 5.* **Sensitivity analysis of loss weights $\lambda_1$ and $\lambda_2$.** We report success rates (%) on three tasks using the DP + DP3 ensemble. Results are averaged over 3 seeds.

| $\lambda_1$ | $\lambda_2$ | Beat Block Hammer | Handover Block | Place Bread Skillet | Avg. |
|---|---|---|---|---|---|
| 0.1 | 0.9 | $88.0\pm2.4$ | $54.0\pm3.2$ | $42.0\pm2.8$ | 61.3 |
| 0.3 | 0.7 | $91.0\pm2.0$ | $60.0\pm2.8$ | $47.0\pm2.4$ | 66.0 |
| 0.5 | 0.5 | $94.0\pm1.6$ | $65.0\pm2.4$ | $51.0\pm2.4$ | 70.0 |
| 0.7 | 0.3 | $\mathbf{97.0}\pm1.0$ | $\mathbf{70.0}\pm2.5$ | $\mathbf{55.0}\pm2.2$ | $\mathbf{74.0}$ |
| 0.9 | 0.1 | $92.0\pm2.0$ | $62.0\pm2.8$ | $48.0\pm2.8$ | 67.3 |

For flow matching, given velocity $\mathbf{v}$ at time $t_f$, Euler integration yields:

$$\Delta\mathbf{x}_f = \mathbf{v} \cdot \Delta\tau, \tag{B.2}$$

where $\Delta\tau = 1/N$ is the step size. Using the unified progress variable $\tau \in [0, 1]$ from Eq. (A.15), we synchronize both models via $t_d = T(1-\tau), t'_d = T(1 - \tau - \Delta\tau)$, and $t_f = \tau$. With both displacements aligned, we compose them as:

$$\mathbf{x}^{(i+1)} = \mathbf{x}^{(i)} + w_d\Delta\mathbf{x}_d + w_f\Delta\mathbf{x}_f, \tag{B.3}$$

where $(w_d, w_f)$ are learnable weights with $w_d + w_f = 1$. This is equivalent to the score/velocity composition in Eq. (A.20)-(A.21), since $\Delta\mathbf{x} \propto v_\theta \cdot \Delta t$. Algorithm 1 summarizes the procedure.

## C. More experiments and details

This section provides a comprehensive set of additional experiments and implementation details to further analyze the behavior and robustness of EnsembleVLA. Specifically, we report (1) inference latency measurements under realistic hardware settings; (2) sensitivity studies on the loss weight hyperparameters $\lambda_1$ and $\lambda_2$; (3) an ablation on the residual correction bound $\delta_{\max}$; and (4) an in-depth analysis of the learned composition weights during training. Together, these results offer insights into the efficiency, stability, and adaptive gate mechanism of our ensemble framework.

### C.1. Inference Latency

We analyze the inference efficiency of EnsembleVLA on an NVIDIA L40S GPU, with results summarized in Table 4. A key advantage of our framework is that the base policies are executed *in parallel*, meaning they do not block each other during inference. Consequently, the ensemble latency (117.5 ms) is only marginally higher than that of the slower individual policy $\pi_{0.5}$ (112.5 ms), with merely 5 ms additional overhead introduced by the composition and refinement modules. Despite an 18% increase in latency compared to $\pi_0$, EnsembleVLA achieves substantial performance gains: on the $\pi_0 + \pi_{0.5}$ combination, the average success rate improves from 65.1% to 74.6%, representing a 9.5% absolute improvement (Table 1). Furthermore, the latency overhead can be effectively mitigated in real-world deployment through action interpolation, which maintains smooth robot motion at higher control frequencies while the ensemble policy computes subsequent actions in the background. These results demonstrate that our ensemble framework achieves a favorable trade-off between computational cost and task performance.

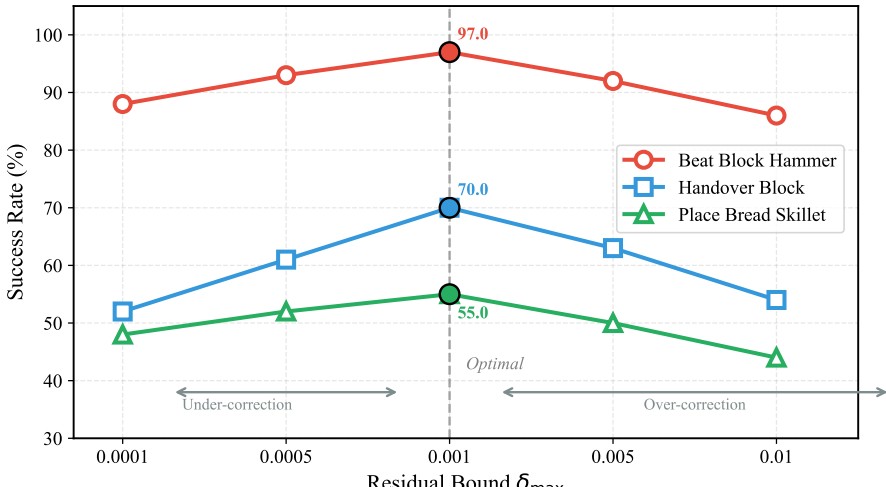

*Figure 5.* **Effect of residual bound $\delta_{\max}$ on success rate.** The optimal value $\delta_{\max} = 0.001$ achieves the best performance across all tasks. Smaller values limit correction capacity, while larger values lead to over-correction.

## C.2. Loss Weight Sensitivity

We investigate the sensitivity of EnsembleVLA to the loss weight hyperparameters $\lambda_1$ (gate supervision) and $\lambda_2$ (delta supervision) in Eq. (23), where $\lambda_1 + \lambda_2 = 1$. Experiments are conducted on the DP + DP3 ensemble across three representative tasks, with results reported in Table 5.

The results reveal several important findings. First, when $\lambda_2$ dominates (for example, $\lambda_1 = 0.1, \lambda_2 = 0.9$), the Gate-Net receives insufficient supervision and fails to learn accurate confidence estimation for selective refinement, leading to degraded performance. Second, when $\lambda_1$ dominates excessively (for example, $\lambda_1 = 0.9, \lambda_2 = 0.1$), the Delta-Net lacks adequate learning signal, resulting in less effective residual corrections. The optimal configuration $\lambda_1 = 0.7$ and $\lambda_2 = 0.3$ achieves the best balance, where the gate supervision is strong enough to guide selective activation while the delta supervision still provides sufficient signal for learning accurate corrections. This suggests that proper confidence-aware gating is slightly more critical than direct residual supervision in our framework. We adopt this setting for all experiments reported in this paper.

## C.3. Residual Bound Sensitivity

We analyze how the residual bound $\delta_{\max}$ affects the performance of EnsembleVLA. Recall that the Delta-Net predicts bounded corrections via $\boldsymbol{\delta} = \delta_{\max} \cdot \tanh(f_\delta(\mathbf{a}))$, where $\delta_{\max}$ controls the maximum allowable correction magnitude. We evaluate different values of $\delta_{\max}$ on the DP + DP3 ensemble across three tasks, with results shown in Figure 5.

The results reveal a clear trade-off in selecting $\delta_{\max}$. When $\delta_{\max}$ is too small (for example, 0.0001), the correction capacity is severely limited, and the Delta-Net cannot sufficiently compensate for errors in the composed actions, resulting in marginal improvements over the base composition. As $\delta_{\max}$ increases to 0.001, the model achieves optimal performance across all three tasks (97.0%, 70.0%, and 55.0% respectively), indicating that this value provides sufficient flexibility for effective refinement. However, when $\delta_{\max}$ becomes excessively large (for example, 0.005 or 0.01), performance degrades noticeably. This is because overly large corrections can destabilize actions that are already near-optimal, introducing unnecessary perturbations that harm execution quality. Based on these findings, we set $\delta_{\max} = 0.001$ as the default configuration, which strikes an effective balance between correction capacity and stability.

## C.4. Composition Weight Analysis

We analyze the evolution of learned composition weights during training to understand how EnsembleVLA balances contributions from different policies. In our framework, the weights $w_{\text{DP}}$ and $w_{\text{DP3}}$ are parameterized via softmax (Eq. (20)) and optimized end-to-end. We conduct experiments on the DP + DP3 ensemble across three representative tasks and

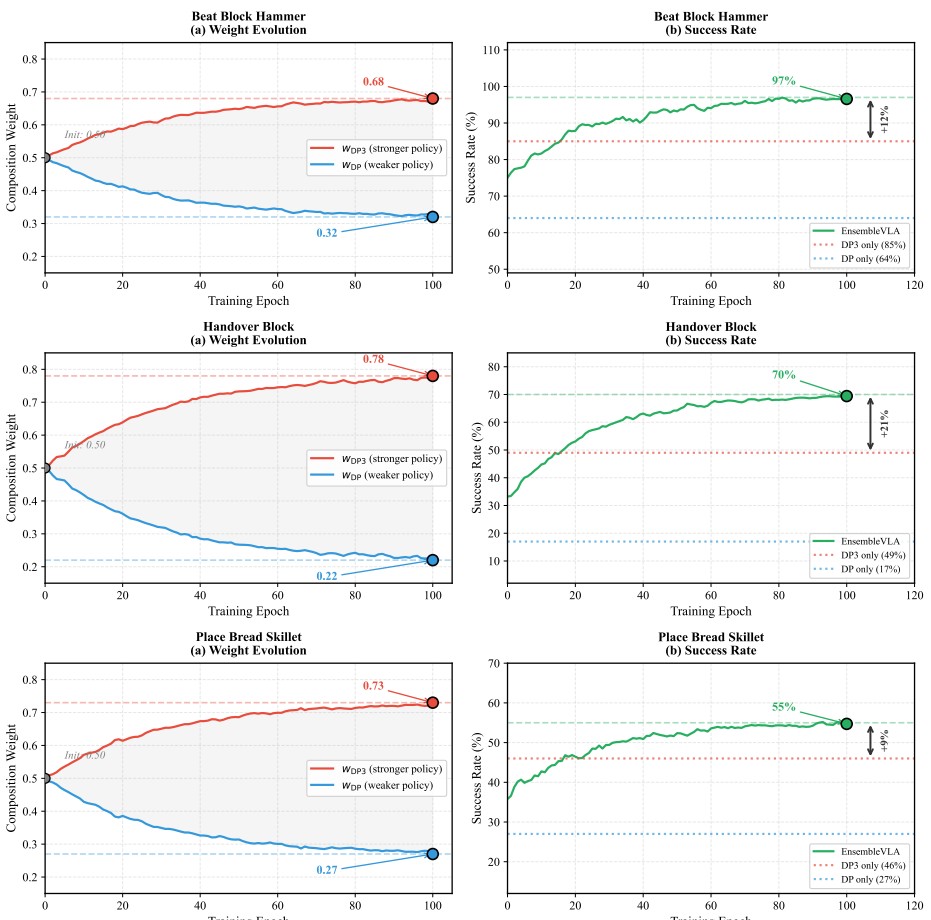

*Figure 6.* **Evolution of composition weights and success rates during training on three tasks.** For each task, the left panel shows the weight dynamics evolving from equal initialization (0.5, 0.5), and the right panel shows the corresponding success rate improvement. The learned weights consistently favor the stronger policy (DP3) while retaining non-trivial contribution from the weaker policy (DP). The final weight allocation varies across tasks, reflecting task-specific policy complementarity.

visualize the weight dynamics and corresponding success rates for each task in Figure 6.

Several interesting patterns emerge from the training dynamics. First, across all three tasks, starting from equal initialization ($w_{DP} = w_{DP3} = 0.5$), the weights gradually diverge as training progresses. The weight assigned to the stronger policy (DP3) steadily increases while the weight for the weaker policy (DP) decreases correspondingly. This adaptive behavior demonstrates that our learnable weight mechanism can automatically identify and prioritize more effective policies. Second, the final weight allocation varies across tasks, reflecting task-specific characteristics. For Handover Block, where the performance gap between DP (17.0%) and DP3 (49.0%) is largest, the learned weights show stronger preference for DP3 ($w_{DP3} \approx 0.78$). In contrast, for Beat Block Hammer, where both policies perform relatively well (DP: 64.0%, DP3: 85.0%), the weight distribution is more balanced ($w_{DP3} \approx 0.68$). Third, notably, the weight of the weaker policy never collapses to zero in any task. This indicates that even a relatively weaker policy provides complementary information that benefits the ensemble, validating the principle that diversity among base learners contributes to overall performance.

## C.5. Experimental Details

**Individual Policy Training.** Before ensemble composition, each base policy is trained independently on task-specific demonstration data. Diffusion Policy (DP) employs a U-Net architecture with a ResNet-18 visual encoder, trained using the DDPM scheduler with 100 diffusion steps. We use the AdamW optimizer with a learning rate of 1e-4, batch size 128, and a cosine learning rate schedule. 3D Diffusion Policy (DP3) follows a similar diffusion framework but operates on point cloud observations, trained with a batch size of 256, a learning rate of 1e-4, and 100 inference steps. $\pi_0$ and $\pi_{0.5}$ are fine-tuned

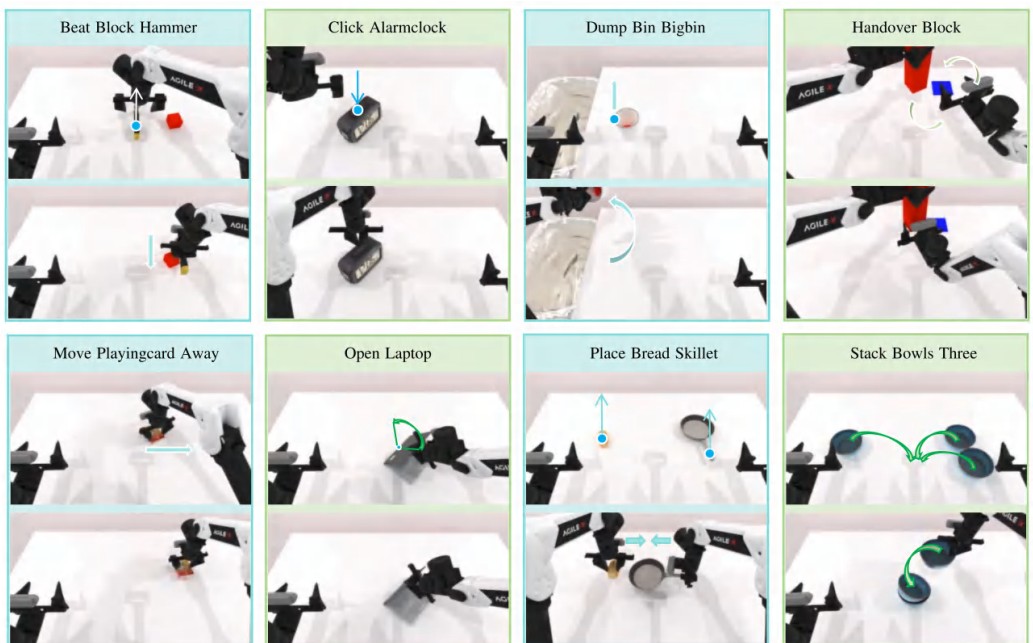

*Figure 7.* Simulated Environment Tasks.

from pre-trained checkpoints using the PI05 framework in RoboTwin2 with JAX/Flax, optimized with AdamW, a batch size of 32, a cosine decay learning rate schedule, and an EMA decay of 0.99. After training, we convert the JAX models to PyTorch using `convert_jax_model_to_pytorch.py` provided in OpenPI and continue training under the same configuration. OpenVLA-OFT is fine-tuned from the OpenVLA-7B foundation model using LoRA (rank 32) and a diffusion (DDPM) action head, with a batch size of 32, a learning rate of 5e-4, and learning rate decay applied at 50,000 steps. All experiments are conducted on NVIDIA H100 80GB GPUs. DP and DP3 are trained on a single GPU, while $\pi_0$, $\pi_{0.5}$, and OpenVLA-OFT require multi-GPU setups due to their larger model sizes.

**Ensemble Policy Training.** We adopt a weight-learning framework to compose the pre-trained policies described above. Training is managed via Hydra-based configuration. All base policy parameters are frozen and only the learnable composition weights and a conservative energy head for action refinement are optimized. The composition weights are parameterized with a softmax to ensure they sum to one. We use the AdamW optimizer with a learning rate of 1e-5 for the residual head and 1e-4 for the composition weights, along with weight decay of 1e-6. The learning rate schedule follows cosine annealing. We set the batch size to 64 and employ 4 data loading workers. All experiments are performed on NVIDIA L40S GPUs using the RoboTwin simulation environment with L515 camera observations. Each task is provided with 100 expert demonstrations. The action space has 14 dimensions, corresponding to the dual-arm ALOHA-AgileX robot configuration.

## D. Simulation Tasks and Description

We conduct our experiments in the RoboTwin 2.0 simulation framework, a scalable and domain-randomized platform designed for robust bimanual robotic manipulation. RoboTwin 2.0 integrates a large-scale object dataset (RoboTwin-OD) with 731 instances across 147 categories, each annotated with semantic and functional affordances. The framework supports automated expert-level trajectory generation via a closed-loop pipeline that combines multimodal large language models (MLLMs) with simulation-in-the-loop feedback. To enhance sim-to-real transfer, RoboTwin 2.0 incorporates systematic domain randomization across five axes: scene clutter, lighting conditions, background textures, tabletop heights, and natural language instructions. The platform also provides embodiment-aware adaptation, enabling the generation of robot-specific manipulation strategies for diverse dual-arm configurations, including Franka, UR5, Piper, ARX-X5, and Aloha-AgileX.

We evaluate our approach on eight representative bimanual manipulation tasks from the RoboTwin 2.0 benchmark: Beat Block Hammer, Click Alarmclock, Open Laptop, Move Playingcard Away, Place Bread Skillet, Dump Bin Bigbin, Handover Block, and Stack Bowls Three. As illustrated in Figure 7, these tasks encompass a diverse set of challenges

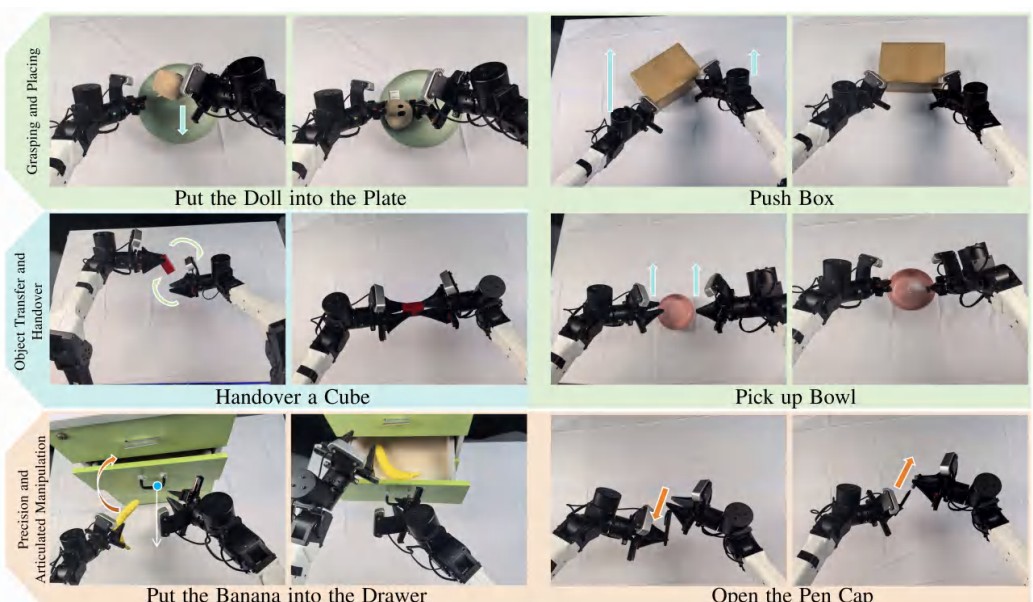

*Figure 8.* Real-World Tasks.

commonly encountered in dual-arm manipulation, including grasping and placing, handover, stacking, tool use, and container manipulation. The selection covers a broad spectrum of task types available in the benchmark, ensuring that our evaluation reflects generalization across different object interactions, motion complexities, and language-conditioned goals. This variety allows us to thoroughly assess policy robustness and adaptability under domain-randomized simulation conditions.

## E. Real-World Setting

### E.1. Hardware Information

We provide a description of the hardware configuration of our real-world experiments. Our proposed EnsembleVLA is deployed on the Cobot Mobile ALOHA platform (Fu et al., 2024), which follows the Mobile ALOHA system architecture. As illustrated in Figure 9, the platform comprises two master arms, two follower arms, two wrist-mounted cameras, and a front-facing camera. Note that in our experiments, we use only one 6-DoF robotic arm and perform stationary manipulation tasks; the robot's mobility features are not utilized.

### E.2. Real-World Task Setup

To evaluate the sim-to-real generalization capability of policies trained in the RoboTwin 2.0 simulation, we design six real-world tabletop tasks that correspond to key skill categories defined in the simulation benchmark. As visualized in Figure 8, these tasks span the following manipulation families:

- **Object Transfer and Handover:** *Handover a cube* requires one arm to grasp and transfer an object to the other arm at a designated handover location, testing sequential coordination and spatial synchronization.

- **Precision and Articulated Manipulation:** *Open the pen cap* involves fine-grained dexterity to rotate and remove a small cap, while *put the banana into the drawer* demands interaction with an articulated object (drawer) under kinematic constraints.

- **Grasping and Placing:** *Pick up bowl* evaluates basic prehensile ability with a curved object. *Push box* examines non-prehensile manipulation via pushing. *Put the doll into the plate* focuses on precise placement with orientation awareness.

Each task is executed 20 times in real-world trials to compute the average success rate, ensuring statistical reliability in performance evaluation.

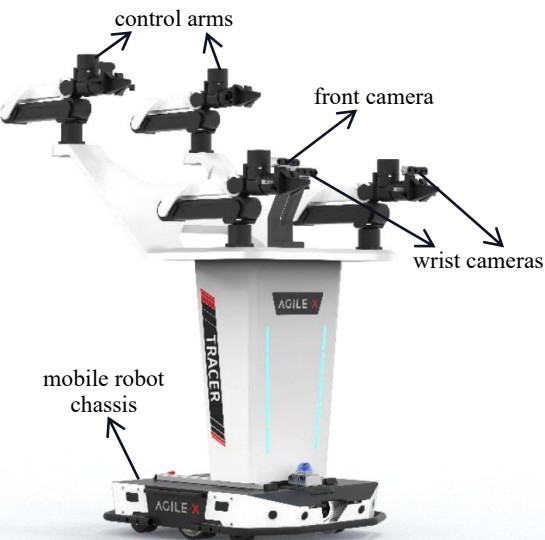

*Figure 9.* Real-World hardware configuration. We use the Cobot Mobile ALOHA, which is equipped with two control arms, two slave arms, two wrist cameras and a frontal camera.

## F. Limitations and Failure case

Our ensemble framework has inherent limitations when composing VLA policies with different generative paradigms. Specifically, when combining flow-based VLA models (e.g., $\pi_0$, $\pi_{0.5}$) with diffusion-based policies (e.g., DP, DP3), we convert the diffusion noise predictions to DDIM-style velocity predictions to align with flow matching's velocity-based formulation. This conversion involves transforming between different normalization spaces and approximating the diffusion denoising direction as a velocity field. Such approximations introduce precision loss, resulting in suboptimal performance compared to composing policies within the same paradigm such as diffusion-diffusion or flow-flow combinations. In real-world deployment, failures primarily arise from hardware constraints such as insufficient gripper friction and joint torque limits, causing grasp slippage or incomplete task execution (e.g., failing to fully open a drawer when placing an banana inside). These mechanical limitations explain the observed sim-to-real performance gap.

## G. Future Work

While this work demonstrates effective ensemble methods for diffusion and flow matching-based VLAs, several promising directions remain for future investigation. First, we aim to extend EnsembleVLA to broader generative model coverage, as our gradient-based ensemble framework naturally generalizes beyond the DP, DP3, openvla-oft, $\pi_0$, and $\pi_{0.5}$ models tested here. Evaluating our approach on more diffusion policy variants and emerging flow matching architectures will further validate the framework's robustness and generality across diverse generative designs. Second, extending our energy-based formulation to non-gradient-based action generation (e.g., direct regression, deterministic feedforward prediction) remains an open challenge; establishing a more general theoretical foundation for composing policies across different paradigms will greatly broaden the applicability of ensemble methods in robotics. Third, we hope this work inspires systematic research on ensemble methods for diverse VLA models in the robotics community: ensemble methods have proven highly effective in machine learning by combining models with distinct inductive biases, and we believe fusing VLAs with different architectures, training objectives, and modeling assumptions can unlock new robotic manipulation capabilities. Key future directions include developing principled model selection strategies, designing adaptive weighting mechanisms, and leveraging ensemble diversity to boost both task performance and real-world deployment robustness.

