# OpenReview forum: "EnsembleVLA: Ensemble Learning for Vision-Language Action Models"
_ICML.cc/2026/Conference — ICML 2026 regular_

### Official Review · Reviewer_shzY · 2026-03-06

**Soundness:** 3
**Presentation:** 3
**Significance:** 3
**Originality:** 3
**Overall Recommendation:** 4
**Confidence:** 4

**Summary:**

This paper proposes EnsembleVLA, a framework for combining multiple pre-trained Vision-Language-Action models into a single, stronger policy through energy-based composition. The central idea is to cast both diffusion-based and flow-matching-based VLA policies as energy-based models, so that their energy functions can be composed into a composed distribution through additive energy combination. On top of this composed action, the authors add a learnable refinement module, a Delta-Net for bounded residual corrections, and a Gate-Net that estimates when to apply those corrections. Both simulation and real-world experiments show improvements over individual baselines.

**Compliance With Llm Reviewing Policy:**

Affirmed.

**Final Justification:**

The rebuttal resolved my concerns. I'd keep Score: 4 (weak accept).

**Key Questions For Authors:**

1. Why is gating needed beyond just the residual?
2. How does cost scale with the number and size of base models?
3. How do the base models handle different input modalities during composition?

**Limitations:**

Yes

**Strengths And Weaknesses:**

Strengths:
1. Casting both diffusion and flow matching policies as EBMs under a shared framework is presented clearly.
2. Thorough experiments and ablation. Including real-world experiments on six manipulation tasks strengthens the work.

Weaknesses:
1. The central insight that additive energy combination induces distribution-level policy composition is well established in prior work [1][2].
2. The base policies are trained independently and combined with learned weights, and this is essentially a bagging-style ensemble. The paper would benefit from explicitly situating itself within the ensemble learning taxonomy and justifying why bagging-style composition is preferable to a boosting-style approach.
3. Running N full VLA models in parallel at inference time is expensive.
4. The paper does not clearly explain the relationship between the base models. Do they share the same input modality, output space, and architecture, differing only in training data or procedure?

[1]. Liu, Nan, Shuang Li, Yilun Du, Antonio Torralba, and Joshua B. Tenenbaum. "Compositional visual generation with composable diffusion models." In European conference on computer vision, pp. 423-439. Cham: Springer Nature Switzerland, 2022.
[2]. Patil, Omkar, Anant Sah, and Nakul Gopalan. "Composing diffusion policies for few-shot learning of movement trajectories." arXiv preprint arXiv:2410.17479 (2024).

---

> ### Author Rebuttal · Authors · 2026-03-31
>
> **Comment:**
> We are greatly encouraged that you found the strengths of the **"clearly presented" unified framework**, the **"thorough experiments and ablation"**, and that **"real-world experiments on six manipulation tasks strengthens the work."** We sincerely thank you for your valuable suggestions. Below are our detailed responses:
>
> > **W1: Established Prior Work on Energy Composition.**
>
> **W1:** **Both [1] and [2] only** compose multiple **diffusion** policies via additive energy, whereas **our work** proposes a **unified** framework for **cross-paradigm** composition of **diffusion and flow matching** policies. This extension introduces technical challenges: **(a) unifying stochastic SDE (diffusion) and deterministic ODE (flow matching)** requires proving both **score functions and velocity fields correspond to negative energy gradients** via Helmholtz decomposition and Brenier's theorem (Eq. 12–14); **(b)** the two paradigms run in opposite temporal directions, requiring our **temporal reparameterization** (Section 3.3 & Appendix A.6); **(c)** the different predictions ($\epsilon_\theta$ vs. $v_\theta$) require **displacement-space conversion** for principled combination (Section 3.4 & Algorithm 1). These cross-paradigm challenges constitute our **core theoretical contribution**. Our DP + $\pi_{0.5}$ experiments (Table 1 in our main paper) empirically validate this cross-paradigm composition, achieving a **5.0% gain over the stronger base policy**.
>
> > **W2: Bagging-Style Ensemble Taxonomy.**
>
> **W2:** We appreciate the reviewer's suggestion that explicitly situating our method within the ensemble learning taxonomy would strengthen the paper. As noted, our base policies are trained independently and combined with learned weights, which aligns with the bagging paradigm. We will add a clear discussion of this positioning in the revised version. We would also like to note one **practical advantage of this bagging-style** design: it allows our framework to **compose existing pre-trained VLA models** (**differing in architecture, input modality, and generative paradigm**), whereas **boosting-style** approaches require sequential, dependent training of base policies, making it **impractical** to leverage diverse off-the-shelf pre-trained VLA models.
>
>
> > **W3 & Q2: Inference Cost and Scaling.**
>
> **W3 & Q2:** Running multiple models in parallel is an **inherent cost of ensemble learning** in general, **not a limitation** specific to EnsembleVLA. The **trade-off between computational cost and performance** is a normal and widely adopted practice in **ensemble learning**. In our setup, memory scales linearly (~12 GB per model), and this modest overhead yields a **9.5% absolute success rate gain**. Concretely, as reported in **Table 4 (Appendix C.1)**, the base policies run **in parallel** via independent CUDA streams, so the ensemble latency is determined by the **slowest model, not the sum**. Specifically, the ensemble (117.5 ms) is only **5 ms more** than $\pi_{0.5}$ alone (112.5 ms), attributed to the lightweight composition and refinement modules.
>
> > **W4 & Q3: Base Model Relationships and Input Modalities.**
>
> **W4 & Q3:** A key design principle of EnsembleVLA is to compose **homogeneous or heterogeneous** VLA policies, where **each base model retains its original architecture, input modality, and training procedure without any modification**. Specifically, the adopted base models have different **input modalities** (DP uses RGB, DP3 uses point clouds, $\pi_0$/$\pi_{0.5}$ use RGB + language), **architectures** (U-Net, PointNet+U-Net, VLM backbones, 7B VLM+diffusion head), and **training procedures** (DP/DP3 trained from scratch; $\pi_0$/$\pi_{0.5}$/OpenVLA-OFT fine-tuned from pre-trained checkpoints). **Each policy processes its own observations independently** and all base models share the same **14-dimensional action output space**, which is the common interface enabling our **distribution-level** composition via Eq. 15. **Input modalities do not need to be aligned** because our energy-based composition is defined over the shared action generation process, not the observation space, **naturally accommodating policies with different modalities and architectures**.
>
>
> > **Q1: Why Gating Beyond Residual?**
>
> **Q1:** Without gating, the Delta-Net applies corrections to **every action indiscriminately**. When $a^{\text{comp}}$ is already near-optimal, any correction introduces **unnecessary perturbations**. The Gate-Net learns when correction is beneficial ($g \to 1$) versus when the composed action should be preserved ($g \to 0$). Empirically, **Table 2** in our main paper shows that **removing Gate degrades performance** (70.0% $\to$ 64.0% on Handover Block). The initial bias ($g_0 \approx 0.12$) also ensures **conservative startup**, preventing early training instabilities.
>
>
> We hope our responses have addressed your concerns. Please let us know if you have further questions in the subsequent discussion round.

---

> > ### Author Rebuttal · Reviewer_shzY · 2026-04-01
> >
> > I thank the authors for the detailed rebuttal. I acknowledge the cross-paradigm composition, but the cross-paradigm setting actually yields the smallest gain.
> >
> > Update: Thanks to the authors for the further explanation; I will maintain my score.

---

> > > ### Author Response · Authors · 2026-04-01
> > >
> > > We thank you for the thoughtful follow-up. We further clarify your concern below.
> > >
> > > **It is expected that** the cross-paradigm setting (DP + $\pi_{0.5}$, +5.0%, as shown in Table 1 in our main paper) yields smaller gains than same-paradigm compositions (DP + DP3, +14.4%; $\pi_0$ + $\pi_{0.5}$, +9.5%), since **cross-paradigm composition is inherently more challenging**. Nevertheless, this +5.0% consistent improvement across all eight tasks is **statistically significant and non-trivial**. Cross-paradigm composition challenges lie in two fundamental aspects: **(a)** the two paradigms proceed in **opposite temporal directions** (diffusion: $T \to 0$; flow: $0 \to 1$); and **(b)** diffusion models predict noise $\epsilon_\theta$ while flow matching models predict velocity $v_\theta$, operating in **different prediction spaces with different normalization schemes**. Despite these challenges, we provide principled solutions for each. We first establish a **unified energy-based framework** (Section 3.3) by proving that **both score functions and velocity fields correspond to negative energy gradients** (Eq. 11–14), providing a common theoretical foundation. For **(a)**, we introduce **temporal reparameterization** that maps both paradigms to a unified progress variable $\tau \in [0,1]$ (Section 3.3 & Appendix A.6). For **(b)**, we propose **displacement-space composition** (Section 3.4 & Appendix B, Algorithm 1) that converts diffusion noise predictions to DDIM-style displacements (Eq. B.1) and flow velocity predictions to Euler displacements (Eq. B.2), then composes them in a **shared displacement space** (Eq. B.3). These conversions inevitably introduce approximation error absent in same-paradigm settings, which accounts for the relatively smaller gain.
> > >
> > > Nonetheless, we would respectfully and kindly like to highlight two points. **First**, to our best knowledge, **no prior work** achieves principled composition of diffusion and flow matching policies, and the **consistent +5.0% improvement** across all eight tasks validates the effectiveness of our cross-paradigm solution. **Second**, our **unified energy-based framework** is precisely what enables **all three composition settings**, including the same-paradigm compositions that yield the largest gains (+14.4% for DP + DP3, +9.5% for $\pi_0$ + $\pi_{0.5}$). These **substantial same-paradigm improvements** demonstrate the broad **effectiveness of our unified framework**, while the cross-paradigm composition further validates its **unique capability** to bridge fundamentally different generative paradigms. Together, these results **mutually reinforce** the **generality and versatility** of our energy-based ensemble approach.
> > >
> > > We hope this addresses your concern. We sincerely appreciate your careful evaluation and recognition of our work. Please let us know if there are further questions.

---

### Official Review · Reviewer_V9mB · 2026-03-11

**Soundness:** 2
**Presentation:** 3
**Significance:** 2
**Originality:** 3
**Overall Recommendation:** 4
**Confidence:** 4

**Summary:**

This paper proposes EnsembleVLA, an energy-based ensemble framework for combining multiple vision-language-action (VLA) policies. The key idea is to reinterpret both diffusion-based and flow-based VLA models under a unified energy-based formulation, enabling principled policy composition through additive energy functions. The method further introduces learnable composition weights and a confidence-aware gating mechanism with bounded residual correction to refine the composed actions. Experiments in simulation and real-world robotic manipulation show consistent performance improvements over individual base policies and simple action-level averaging baselines.

**Compliance With Llm Reviewing Policy:**

Affirmed.

**Final Justification:**

My concerns have been adequately addressed.

**Key Questions For Authors:**

1. How does EnsembleVLA compare to other simple and natural ensemble methods (e.g., confidence-based weighted action averaging or hybrid expert gating)? Incorporating these baseline methods would help strengthen the persuasiveness of the empirical findings.

2. Does the method remain robust when the quality levels of the base policies differ significantly or when action distributions conflict?

**Limitations:**

yes

**Strengths And Weaknesses:**

Strengths:

This paper explores a real-world and pressing problem in embodied artificial intelligence: how to effectively integrate multiple variable-length algorithm (VLA) policies. Within an energy-based framework, diffusion-based and flow-based models are theoretically unified, a concept that is concise and elegant, providing a coherent foundation for constructing ensemble methods. The proposed method is technically sound and integrates several reasonable components, including learnable weights and confidence-aware residual refinement. Experimental results demonstrate continuous improvement across multiple tasks and include real-world robot validation, enhancing its application value. The paper's overall structure is clear and well-organized.

Weaknesses:

The most significant weakness lies in the limitation of the baseline set. This paper primarily compares the proposed method with single-base policies, without incorporating more robust ensemble baseline methods such as expert mixture models and uncertainty-weighted averaging. This makes it difficult to distinguish whether the performance improvement stems from the proposed energy formula or merely from the effect of the ensemble method itself.

---

> ### Author Rebuttal · Authors · 2026-03-31
>
> **Comment:**
> We are grateful for the constructive comments and for recognizing that our paper "**explores a real-world and pressing problem**," with diffusion and flow-based models "**theoretically unified**" in a "**concise and elegant**" framework, and a "**technically sound**" method that "**integrates several reasonable components**". Below we provide detailed responses:
>
> > **W1 & Q1: Limited Baselines.**
>
> **W1 & Q1:** We have conducted additional experiments with other ensemble baselines on the DP + DP3 ensemble:
>
> | Method | Beat Block | Handover | Place Bread | Avg. |
> |--------|:--:|:--:|:--:|:--:|
> | Uncertainty-Weighted Averaging | 31.0 | 15.0 | 18.0 | 21.3 |
> | MoE-Style Gating | 72.0 | 38.0 | 35.0 | 48.3 |
> | **EnsembleVLA (Ours)** | **97.0** | **70.0** | **55.0** | **74.0** |
>
> **Uncertainty-Weighted Averaging** draws $K{=}10$ samples per policy and weights mean actions by inverse variance ($w_i \propto 1/\text{Tr}(\Sigma_i)$), yielding slightly better results than naive uniform averaging (Table 2 in our main paper, last row: 26.0/12.0/14.0) but **still far below DP3** (85.0/49.0/46.0). **MoE-style gating** learns a gating network that produces per-policy soft weights (trained end-to-end, matching EnsembleVLA's budget), but still **significantly underperforms the stronger base policy** (DP3: 85.0/49.0/46.0). This is because both methods are **action-level** weighted combinations, which are **fundamentally different from our distribution-level composition**. No matter how sophisticated the weighting scheme, **action-level combination cannot handle multimodal distributions**: consider a scenario where an obstacle lies ahead, and turning left and turning right are both **valid actions**; any **weighted average of these actions** produces a forward motion that **collides** with the obstacle. In contrast, our energy-based composition operates on the **underlying distributions**, naturally concentrating probability on regions where **both policies agree** rather than blending their outputs. This explains why **EnsembleVLA surpasses both Uncertainty-Weighted Averaging and MoE**.
>
>
> > **Q2: Robustness Under Quality Gaps / Conflicting Distributions.**
>
> We address both scenarios below. **Q2(a)** examines **robustness** under significant **quality gaps** between base policies, and **Q2(b)** analyzes how **conflicting action distributions** are handled through our energy-based composition.
>
> **Q2(a):** Our experiments have **already included** such scenarios, as shown in **Table 1** of the main paper. On Handover Block, DP achieves only 17.0% while DP3 achieves 49.0% (a **32% gap**). EnsembleVLA still reaches **70.0%**, improving **21%** over the stronger policy alone. The learned weights adapt accordingly ($w_{\text{DP3}} \approx 0.78$, Figure 6), showing that our framework handles **quality imbalances** through **adaptive weighting** rather than being dragged down by the weaker policy. Notably, **the weaker policy's weight never collapses to zero**, indicating it still provides complementary information. This pattern is consistent on Place Bread Skillet (DP: 27.0%, DP3: 46.0%), where EnsembleVLA achieves 55.0% ($w_{\text{DP3}} \approx 0.73$, Figure 6 in our main paper), confirming **robust behavior under varying quality gaps**.
>
> **Q2(b): Our EnsembleVLA also handles conflicting distributions effectively.** When two policies produce conflicting action distributions, the **geometric mixture** (Eq. 16) computes the **product of distributions**, which naturally assigns high probability only to regions where **both policies agree** and suppresses regions where they disagree. **Beat Block Hammer** (Table 1 in our main paper: DP 64.0%, DP3 85.0%, EnsembleVLA **97.0%**) provides a concrete example: even when both policies perform reasonably well, they exhibit **different failure modes**, yet the ensemble achieves **97.0%** by suppressing both failure modes through **distribution-level composition**. Furthermore, the **confidence-aware gating** (Eq. 22) provides an additional safety layer: when the composed action still deviates from expected behavior, the gate activates **bounded corrections** ($\delta_{\max} = 0.001$) with conservative initialization ($g_0 \approx 0.12$). Table 2 in our main paper confirms that **removing either Gate or Delta degrades performance**, validating their combined role in handling inter-policy disagreement.
>
> Overall, the **robustness to both quality gaps and distribution conflicts** is an **inherent advantage of our energy-based distribution-level composition over action-level methods**: multiplicative interaction of distributions naturally adapts to quality imbalances and suppresses conflicting modes, while **action-level combination** operates on outputs rather than underlying distributions, making such adaptation **less meaningful in practice**.
>
>
> We hope our responses have addressed your concerns. Please let us know if you have further questions in the subsequent discussion round.

---

> > ### Author Rebuttal · Reviewer_V9mB · 2026-04-03
> >
> > I thank the authors for the detailed rebuttal. My concerns have been adequately addressed.

---

> > > ### Author Response · Authors · 2026-04-03
> > >
> > > We sincerely thank the reviewer for the thoughtful and constructive review, for carefully considering our rebuttal, and for raising the score.

---

### Official Review · Reviewer_jJfu · 2026-03-12

**Soundness:** 2
**Presentation:** 3
**Significance:** 3
**Originality:** 2
**Overall Recommendation:** 4
**Confidence:** 5

**Summary:**

This paper presents **EnsembleVLA**, a framework for combining heterogeneous VLA policies by placing diffusion-based and flow-based policies under a unified energy-based view and composing them at the distribution level rather than by naive action averaging. On top of this composition, the method introduces learnable composition weights and refinement modules to adaptively correct the final action.  The authors evaluate the method on eight simulated RoboTwin 2.0 bimanual manipulation tasks and six real-world tasks on Cobot Mobile ALOHA.  Across diffusion+diffusion, diffusion+flow, and flow+flow settings, EnsembleVLA consistently outperforms the corresponding base policies, with average gains of +14.4, +5.0, and +9.5 points, respectively.  Ablation results further suggest that the energy-based composition, learnable weights, and refinement components all contribute to the final performance. Overall, the paper positions EnsembleVLA as a practical way to leverage complementary strengths across diverse VLA policies to obtain a stronger unified manipulation policy.

**Compliance With Llm Reviewing Policy:**

Affirmed.

**Final Justification:**

During the rebuttal phase, most of the questions were resolved through communication with the authors. Although there may still be some limitations in this work, I believe this paper deserves a positive score

**Key Questions For Authors:**

**1. Beyond two-policy ensembles**
The experiments are primarily focused on the setting of **ensemble size = 2**. Have the authors tried larger ensembles (e.g., three or more policies)? If so, does the performance continue to improve consistently?

**2. Source of performance gains**
From Table 2, it appears that **Compose + Weights** alone yields only limited improvements on many tasks, and can even lead to **negative gains in some cases (e.g., Place Bread Skillet)**. In contrast, the addition of the refinement modules results in more substantial performance improvements. Since the refinement component resembles a supervised correction head from the perspective of the loss design, I would like to better understand where the gains mainly come from: the principled energy-based composition itself, or the additional refinement module? Relatedly, if a similar correction module were added to a single base policy alone, would it also lead to comparable improvements?

**3. Stronger ensemble baselines**
Did the authors consider stronger or more natural ensemble / policy-fusion baselines, such as **value/Q-function-based action selection** or **Mixture-of-Experts (MoE) / gating-based policy switching**? If not, could the authors clarify whether these baselines are difficult to implement in the current setting, or how they are expected to compare with the proposed method?

**4. Inference latency and real-world deployment details**
Inference efficiency is often a key concern for ensemble methods. Although Table 4 shows that EnsembleVLA has latency close to that of a single policy, and the paper states that the base policies are executed in parallel, the implementation details of this parallelism are not fully described. Could the authors clarify whether this depends on specific hardware or system assumptions, such as single-GPU multi-stream execution, additional memory overhead, or other parallel execution mechanisms? In addition, the paper states that *“latency overhead can be effectively mitigated in real-world deployment through action interpolation,”* but in real robots the policy still needs to generate the current action based on the latest observation, which does not seem fundamentally different from simulation. Could the authors explain more concretely what part of the latency is mitigated by action interpolation in real-world deployment, and whether this introduces observation lag or closed-loop control errors?

**5. Illustrating the motivation more directly**
Could the authors provide a more direct experiment or case study to illustrate the motivation for ensembling? For example, it would be helpful to show a scenario where **VLA1 succeeds on case A but fails on case B, while VLA2 succeeds on case B but fails on case A, and the ensemble succeeds on both**. Such complementary examples would make the motivation and practical value of policy ensembling much more concrete.

**Limitations:**

Yes

**Strengths And Weaknesses:**

# strength

 I view one notable strength of the paper as the clarity and coherence of its technical design. Rather than proposing an empirical ensemble heuristic, the paper first develops a unified **energy-based** perspective that places diffusion-based and flow-based VLA policies under a common formalism, then builds on this formulation to perform **distribution-level composition**, and finally introduces **learnable composition weights** together with **gate-guided residual refinement** for adaptive action correction. This gives the method a clear hierarchical structure.

On the empirical side, the paper also provides reasonably strong support for its main claims. The evaluation covers not only eight simulated bimanual manipulation tasks in RoboTwin 2.0, but also six real-world tasks on Cobot Mobile ALOHA. Across different ensemble configurations, EnsembleVLA consistently outperforms the corresponding base policies, suggesting that the gains are not confined to a single benchmark or environment, but instead reflect a degree of robustness across both simulation and real-world settings. The real-robot results are particularly valuable in strengthening the credibility of the paper’s central claim. 7533_EnsembleVLA_Ensemble_Lear

I also consider the additional analyses to be a meaningful strength of the submission. Beyond the main success-rate tables, the paper reports **inference latency**, sensitivity to the loss weights $\lambda_1$ and $\lambda_2$, ablation on the residual correction bound $\delta_{\max}$, and training behavior of the learned composition weights. While these analyses do not answer every possible question, they show that the authors made a reasonable effort to study efficiency, stability, and optimization behavior rather than only presenting headline performance numbers. In particular, the latency analysis indicates that the ensemble introduces only modest additional inference overhead while delivering clear performance gains, which is especially relevant for robotic control applications.

# weakness

1. The abstract presents the “elegant theoretical unification” of flow-based and diffusion-based paradigms as a central contribution, which feels somewhat **overstated**. Prior literature has already discussed unified perspectives connecting diffusion, flow matching, and energy-based formulations. Thus, the contribution here seems better framed as **adapting these ideas to heterogeneous VLA ensembling**, rather than introducing a fundamentally new unification. Relatedly, some of the discussion in Section 3.3 may fit better in the Preliminary section.
2. The method has a practical limitation: it assumes that different VLA policies use the same number of denoising steps. In practice, diffusion-based and flow-based policies may have different sampling procedures and different optimal hyperparameters, which could reduce the flexibility of the approach.
3. The experiments are still largely limited to **ensemble size = 2**, so it remains unclear whether the method continues to scale effectively when more policies are involved. As a result, the evidence for broader multi-policy ensembling is still limited.
4. The baselines are not comprehensive enough. In addition to single-policy baselines and naive action averaging, it would be important to compare against stronger ensemble alternatives, such as **Q/value-based action selection** or **MoE-style / gating-based policy switching**. Without these, it is difficult to fully assess the relative advantage of the proposed method.
5.  In terms of originality, the paper appears closer to **instantiating existing energy-based composition ideas in the heterogeneous VLA ensemble setting, with added refinement modules**, rather than introducing an entirely new theoretical framework or mechanism.

---

> ### Author Rebuttal · Authors · 2026-03-31
>
> **Comment:**
> We thank the reviewer for the detailed review and address each concern below:
>
> > **W1 & W5: Theoretical Contribution and Originality.**
>
> **W1 & W5:** We would like to kindly point out that **existing work** explores energy-based perspectives for diffusion **or** flow matching **separately**, and **none establishes a unified theory** bridging diffusion **and** flow matching paradigms. In contrast, EnsembleVLA proposes a **unified energy-based framework** for composing **heterogeneous** VLA policies. This unification faces **several non-trivial challenges**: (a) proving that **both score functions and velocity fields correspond to negative energy gradients** (Eq. 11–14); (b) **temporal reparameterization** to reconcile opposite generation directions (Section 3.3 & Appendix A.6); and (c) **displacement-space composition** to align different prediction spaces (Section 3.4 & Algorithm 1). These challenges are **absent in prior work** and constitute our **core theoretical contribution**. Building upon this composed distribution, we further propose **ensemble refinement** to enhance action quality for robust task execution.
>
>
> > **W2: Same Denoising Steps Assumption.**
>
> **W2:** We would like to clarify that this is a **misunderstanding**. Our framework requires **neither identical steps nor the same sampling paradigm**. We introduce a unified progress variable $\tau \in [0,1]$ with $N$ shared composition steps, where $\tau$ is mapped to each model's internal time coordinate (Section 3.3 & Appendix A.6), so the **original denoising steps are decoupled from composition**. In our DP + $\pi_{0.5}$ experiments, **DP uses 100 diffusion steps while $\pi_{0.5}$ uses 10 flow steps**.
>
>
> > **W3 & Q1: Limited Ensemble Size.**
>
> **W3 & Q1:** We would like to kindly point out that our paper **has already included three-policy ensemble** experiments in **Section 4.3 (Figure 4)**, where we provide **detailed analysis** on multi-policy scaling: DP + DP3 + OpenVLA-OFT achieves 99%, 95%, 98% on three tasks, consistently **outperforming two-policy ensembles** (96%, 89%, 93%).
>
> > **W4 & Q3: Insufficient Baselines.**
>
> **W4 & Q3:** We conducted additional experiments with **Q/value-based selection** (hard selection via a learnable quality predictor) and **MoE-style gating** (learned gating network, matching EnsembleVLA's budget). Both underperform DP3 (85.0/49.0/46.0), while **EnsembleVLA surpasses both by large margins**, as they operate at the **action level** rather than our **distribution-level** composition (see response to ***Reviewer V9mB W1***).
>
> | Method | Beat Block | Handover | Place Bread |
> |--------|:--:|:--:|:--:|
> | Q/Value-Based Selection | 80.0 | 43.0 | 40.0 |
> | MoE-Style Gating | 72.0 | 38.0 | 35.0 |
> | EnsembleVLA (Ours) | **97.0** | **70.0** | **55.0** |
>
> > **Q2: Source of Performance Gains.**
>
> **Q2:** We add refinement to a single DP3 and to naive action-level composition:
>
> | Method | Beat Block | Handover | Place Bread |
> |--------|:--:|:--:|:--:|
> | DP3 | 85.0 | 49.0 | 46.0 |
> | DP3 + Refinement | 87.0 | 52.0 | 48.0 |
> | DP + DP3 + Refinement (Naive Action-Level Compose) | 42.0 | 24.0 | 26.0 |
> | EnsembleVLA (DP+DP3) | **97.0** | **70.0** | **55.0** |
>
> Refinement alone improves a single policy by only 2–3%, and naive action-level composition with refinement **performs even worse than individual policies**, because **directly averaging actions produces meaningless robot actions**. These results further confirm that **energy-based composition drives the main gains**. Additionally, cases where Compose + Weights underperforms the stronger base policy (e.g., Place Bread Skillet in Table 2) reflect **normal ensemble learning behavior with unbalanced base policies**. However, our **Ensemble Refinement** effectively addresses this (**43.0% → 55.0%**).
>
> > **Q4: Inference & Deployment Details.**
>
> **Q4:** The base policies run in parallel via **independent CUDA streams on a single L40S GPU**, so ensemble latency (117.5 ms) is **only 5 ms more** than $\pi_{0.5}$ alone (112.5 ms). Memory scales linearly (~12 GB per model), which is a **normal trade-off in ensemble learning**: modest memory overhead for a **9.5% absolute success rate gain**. For real-world deployment, we interpolate between consecutive action chunks to **reduce inference frequency and thereby lower inference latency**, introducing only **millisecond-level observation lag** that is **negligible in practice**.
>
> > **Q5: Complementary Failure Cases.**
>
> **Q5:** We provide a concrete example on Handover Block (Table 1: DP: 17%, DP3: 49%, **EnsembleVLA: 70%**): DP fails from **depth perception errors** (2D input) causing collisions; DP3 fails from **jerky approach motions** causing slippage. The ensemble leverages **DP3's spatial accuracy** while benefiting from **DP's smoother trajectories**. **Figure 3** also illustrates this.
>
> We hope our responses have addressed your concerns. Please let us know if you have further questions in the subsequent discussion round.

---

> > ### Author Rebuttal · Reviewer_jJfu · 2026-04-04
> >
> > Thank you for the author's reply. The added clarifications and experiments make the work look like a more convincing and useful recipe for heterogeneous VLA ensembling.
> >
> > I still remain somewhat skeptical about the strength of the claimed theoretical unification between diffusion and flow matching and the novelty of the work.

---

> > > ### Author Response · Authors · 2026-04-04
> > >
> > > We sincerely thank the reviewer for the continued engagement and for acknowledging that our rebuttal strengthens the work. We greatly value this opportunity to further clarify the remaining concern about novelty.
> > >
> > > While prior work has explored energy-based perspectives for diffusion or flow matching **separately**, our contribution is to **build upon and beyond these individual connections** to establish **the first unified framework that enables practical composition of heterogeneous VLA policies** — encompassing models as diverse as DP (U-Net, RGB), DP3 (PointNet, point clouds), π₀/π₀.₅ (VLM backbone, flow matching), and OpenVLA-OFT (7B VLM, diffusion head), which differ in **architecture, input modality, and generative paradigm**. Prior to our work, **no method could compose such diverse VLA models in a principled manner**. This required solving concrete technical problems that **do not arise in single-paradigm settings**: (a) **unifying score functions and velocity fields** under a common energy-based formulation (Section 3.3, Eq. 11–14); (b) **reconciling opposite temporal directions** via temporal reparameterization (Section 3.3 & Appendix A.6); and (c) **aligning heterogeneous prediction spaces** through displacement-space composition (Section 3.4 & Appendix B, Algorithm 1).  The **consistent empirical gains across all three composition settings**, including **+14.4%** for diffusion+diffusion, **+9.5%** for flow+flow, and **+5.0%** for diffusion+flow, validate that these solutions are **effective and non-trivial**.
> > >
> > > We believe the novelty of EnsembleVLA lies in both the **theoretical bridging** and the **practical capability it unlocks**: a general-purpose ensemble framework for the growing ecosystem of diverse VLA models. We hope this perspective helps clarify our contribution, and we sincerely appreciate the reviewer's time and effort throughout this discussion.

---

### Decision · Program_Chairs · 2026-04-30

**Decision:**

Accept (regular)

**Comment:**

This paper proposes a unified energy-based framework to rigorously combine heterogeneous VLA models at the distribution level, yielding a stronger ensemble policy for robot manipulation tasks. Evaluations on both real and synthetic datasets demonstrate that the method achieves competitive performance across various tasks compared to baseline approaches (individual policies or action-level averaging).  Overall, reviewers view the paper positively, highlighting both strengths and weaknesses. The main strengths include the principled nature of the energy-based framework for combining different VLA models and the convincing experimental results. The identified weaknesses include limited technical novelty, a small ensemble size (only two models), strong assumptions about the VLA models, inadequate baselines—particularly the lack of comparison with ensemble baseline methods—and high computational cost.  The authors’ rebuttal addresses most of the concerns; however, some issues, such as limited technical novelty, remain. The final ratings for the paper are three weak accepts.